# Ubiquitous MEIS transcription factors actuate lineage-specific transcription to establish cell fate

Zoulfia Darieva[1], Peyman Zarrineh[1], Naomi Phillips [ID][1,3], Joshua Mallen[1,3], Araceli Garcia Mora [ID][1,3], Ian Donaldson [ID][1], Laure Bridoux[1], Megan Douglas [ID][1], Sara F Dias Henriques[1], Dorothea Schulte[2], Matthew J Birket [ID][1✉] & Nicoletta Bobola [ID][1✉]

## Abstract

**Control of gene expression is commonly mediated by distinct combinations of transcription factors (TFs). This cooperative action allows the integration of multiple biological signals at regulatory elements, resulting in highly specific gene expression patterns. It is unclear whether combinatorial binding is also necessary to bring together TFs with distinct biochemical functions, which collaborate to effectively recruit and activate RNA polymerase II. Using a cardiac differentiation model, we find that the largely ubiquitous homeodomain proteins MEIS act as actuators, fully activating transcriptional programs selected by lineage-restricted TFs. Combinatorial binding of MEIS with lineage-enriched TFs, GATA, and HOX, provides selectivity, guiding MEIS to function at cardiac-specific enhancers. In turn, MEIS TFs promote the accumulation of the methyltransferase KMT2D to initiate lineage-specific enhancer commissioning. MEIS combinatorial binding dynamics, dictated by the changing dosage of its partners, drive cells into progressive stages of differentiation. Our results uncover tissue-specific transcriptional activation as the result of ubiquitous actuator TFs harnessing general transcriptional activator at tissue-specific enhancers, to which they are directed by binding with lineage- and domain-specific TFs.**

**Keywords** Cardiac Differentiation; Combinatorial Binding; Transcription Factors

**Subject Categories** Cardiovascular System; Chromatin, Transcription & Genomics; Development

## Introduction

Control of gene expression determines and maintains cellular identity and function in embryonic development and adult tissue homeostasis. Lineage-specific TFs orchestrate the precise gene expression programs that determine cell fate (Spitz and Furlong,

2012). However, with rare exceptions, lineage-specific TFs are unable to efficiently program cells into their specific cell type (Wang et al, 2021). Examples from animal development or cellular reprogramming indicate that transcriptional regulation is commonly mediated by distinct combinations of TFs (Reiter et al, 2017). Combinatorial binding enables the integration of multiple biological inputs at *cis*-regulatory elements to generate highly specific regulatory outcomes in space and time (Spitz and Furlong, 2012). This collaborative binding is favored because multiple TFs, interacting with DNA simultaneously, display higher affinity for their designated sites (Mirny, 2010; Moyle-Heyrman et al, 2011). The role of TF cooperativity in stabilizing TF–DNA binding is well-established. However, our understanding of how these TF combinations regulate RNA polymerase II activity to affect gene expression remains incomplete. The biochemical effects of TFs after binding DNA are largely unmapped. Specifically, it is unclear if DNA-bound TFs recruit co-activators equivalently, making them interchangeable, or if combinatorial binding assembles TF complexes with emerging functional properties crucial for transcriptional activation.

MEIS TFs belong to the three amino acid loop extension (TALE) superclass, which includes evolutionary ancient TFs with an atypical homeodomain (Bobola and Sagerström, 2024; Bürglin and Affolter, 2016; Selleri et al, 2019). Initially discovered as HOX partners, MEIS1-2 TFs are essential for the development of many organs and can function as oncogenes (Schulte and Geerts, 2019). While largely ubiquitous, MEIS1-2 TFs have been implicated in the control of tissue-specific transcription. In mouse development, high-confidence MEIS binding events occur at tissue-specific locations across different embryonic tissues. These binding events are enriched in motifs recognized by tissue-specific TFs and associated with increased enhancer activity and gene expression in the same tissue (Bridoux et al, 2020a; Phuycharoen et al, 2020). The potential of widely expressed TFs like MEIS to drive the acquisition of tissue-specificity, and the precise mechanisms by which they might achieve this, remain an open question, with implications for cell reprogramming.

Here we find that MEIS proteins act as *actuator TFs*, fully activating the transcriptional programs directed by lineage-specific TFs to orchestrate the dynamic progression of cardiac differentiation. MEIS

[1]Faculty of Biology, Medicine and Health, University of Manchester, Manchester, UK. [2]Goethe University, University Hospital Frankfurt, Neurological Institute (Edinger Institute), Frankfurt am Main, Germany. [3]These authors contributed equally: Naomi Phillips, Joshua Mallen, Araceli Garcia Mora. ✉E-mail: matthew.birket@manchester.ac.uk; nicoletta.bobola@manchester.ac.uk

recognition motifs are enriched in human developmental enhancers active in distinct tissues, heart, brain, limb, kidney and adrenal glands. Using a cardiac differentiation system, we find that MEIS is essential to initiate a cardiac-specific gene expression program. Combinatorial binding of MEIS with GATA and HOX directs MEIS to function at cardiac-specific enhancers. Here, MEIS promotes the accumulation of the methyltransferase KMT2D to initiate lineage-specific enhancer commissioning. MEIS trades partners to orchestrate the dynamic progression of differentiation. Our results uncover tissue-specific transcriptional activation as the result of ubiquitous actuator TFs harnessing general transcriptional activator at tissue-specific enhancers, to which they are directed by binding with lineage- and domain-specific TFs.

## Results

### MEIS1-2 are essential for cardiac lineage differentiation

Using a screen designed to map TF combinatorial binding in human developmental enhancers (Garcia-Mora et al, 2023), we found a significant enrichment of recognition motifs for TALE TFs nearby tissue-specific sequence signatures. TALE motifs occur in the proximity of sequences recognized by tissue-restricted TFs in the heart (ventricle), brain, upper limb, adrenal gland, and kidney enhancers (Fig. 1A). In contrast to their potential co-binding partners, MEIS TFs are broadly expressed (Fig. 1B). Cardiac TFs are the most highly significant predicted partners of MEIS. Therefore, to understand the exact contribution of ubiquitous TFs like MEIS to the acquisition of tissue-specificity, we exploited a well-established 3D model of cardiac differentiation (Fig. 1C) (Birket et al, 2015). Initially, we investigated the expression dynamics of *MEIS* genes within this system. Single-cell analysis of human embryonic stem cells (hESC) differentiation into cardiomyocytes (Fig. 1D) reflects the temporal progression from mesodermal cells, marked by *TBXT* at d3, to *MESP1*-positive cardiac mesoderm (Fig. 1E). The expression of *MEIS1* and *MEIS2* follows cardiac mesoderm induction and coincides with the emergence of cardiac progenitor cells (Fig. 1F). Both *MEIS1* and *MEIS2* remain highly expressed in *MEF2C-TBX5*-positive late progenitors (Fig. 1G). While cardiomyocytes (identified by the expression of *NKX2-5* and *MYH6*) (Fig. 1H) and epicardial cells (*WT1-TBX18*-positive) (Fig. 1I) maintain high *MEIS2* expression, *MEIS1* expression decreases at later, differentiated stages (Fig. 1F,J). Relative to *MEIS1* and *MEIS2*, whose expression is first detected in early progenitors, *MEIS3* is already expressed in the mesoderm and exhibits less pronounced temporal dynamics throughout differentiation (Fig. EV1A). Likewise, *PBX* genes, whose encoded TFs also bind the TALE motif, are expressed consistently across the analyzed stages of differentiation (Fig. EV1B).

*MEIS1-2* expression in early cardiac progenitors is suggestive of a role in cardiac lineage determination. Starting from a NKX2-5-GFP reporter hESC line (Elliott et al, 2011), we used CRISPR–Cas9 to generate a knockout of MEIS1 and MEIS2 (referred to as MEIS KO) (Fig. EV2A). Wild-type embryoid bodies (EBs) became GFP-positive by day (d) 8 of differentiation and contracted soon after, indicating proper cardiomyocyte differentiation; in contrast, MEIS KO EBs remained GFP-negative and failed to contract (Figs. 2A and EV2B). At d12, MEIS KO EBs lacked markers of cardiomyocytes and epicardial cells, indicating a general failure in cardiac progenitor specification or differentiation (Fig. EV2C). Similarly, MEIS KO in MAN13 hESCs consistently showed a lack of contraction, along with failed upregulation of cardiac markers (Fig. EV2D). The absence of MEIS1-2 in hESC did not affect the expression of pluripotency factors or markers of mesoderm differentiation (*TBXT*, *EOMES*, *MESP1*), indicating unaffected initial stages (Fig. 2B). However, dysfunction was evident by the failed upregulation of crucial cardiac TFs like *GATA4* and *NKX2-5* in the absence of MEIS1-2 (Fig. 2B).

To examine the requirement for MEIS1-2 in cardiomyocyte differentiation, we compared transcriptomes of wild-type and MEIS KO (two MEIS KO clones, KO1 and KO2) at two stages, corresponding to early (d5, shortly after the onset of *MEIS1* transcription) and late (d7) cardiac progenitors. We found a total of 457 and 1240 differentially expressed (DE) genes, respectively (log fold change (log FC) > [1]; $P$adj <0.05), at d5 and d7, respectively) (Fig. 2C; Dataset EV1). At the earliest time point, downregulated genes in the MEIS KO represent the largest fraction of dysregulated genes (>70%). By d7, there is a similar ratio of upregulated and downregulated genes. Changes measured close to the start of MEIS expression (d5) likely reflect direct regulation by MEIS, suggesting that MEIS transcription factors predominantly function as transcriptional activators. Genes activated by MEIS (d5 down in MEIS KO) are associated with heart development and cardiac differentiation. In contrast, upregulated genes (d5 up in MEIS KO) are linked to multiple embryonic processes, but not including cardiac development (Fig. 2D). When genes associated with cardiac mesoderm gene ontologies (GO) are selected, genes downregulated in MEIS KO (d5) are enriched in cardiac-specific transcripts, while upregulated genes are generally expressed in the mesoderm (e.g., *TWIST1*, *FOXC1-2*, *SIX1*) (Fig. 2E). At d7, genes downregulated in MEIS KO also include the epicardial markers, *TBX18* and *WT1* (Fig. EV2D). These results indicate that MEIS KO cells fail to activate cardiac-specific genes, possibly remaining in a more primitive mesodermal stage. Consistent with this hypothesis, PCA analysis positions MEIS KO cells between mesoderm and cardiac progenitors in a time course of cardiac differentiation (Data ref: Bertero et al, 2019a; Bertero et al, 2019b), with limited change in MEIS KO cells from d5 to d7 (Fig. 2F). In sum, these findings indicate that MEIS1-2 are essential TFs for cardiac lineage differentiation, highlighting the requirement of ubiquitous TFs in tissue-specific transcription.

### MEIS1-2 function as activators to establish a cardiac-specific program

Physical access to DNA is a dynamic property of chromatin that reflects the *cis*-regulatory control of gene expression. To examine the effects of MEIS on the chromatin landscape, we mapped chromatin accessibility in wild-type and MEIS KO cells and assessed quantitative differences in ATAC-seq signal intensities at early (d5) and late (d12) cardiac differentiation stages. Globally, MEIS1-2 are (directly or indirectly) responsible for a relatively small fraction of chromatin accessibility: more than 70% open chromatin remains unperturbed in MEIS KO cells, and only around 15% regions loose accessibility and an equal fraction gains accessibility (Fig. 3A). Strikingly, however, the fraction of regions whose gain in accessibility depends on MEIS (ATAC WT > KO), is specifically associated with cardiovascular development and differentiation (Fig. 3B). Chromatin accessibility shows distinct temporal dynamics: at d5, higher accessibility is largely

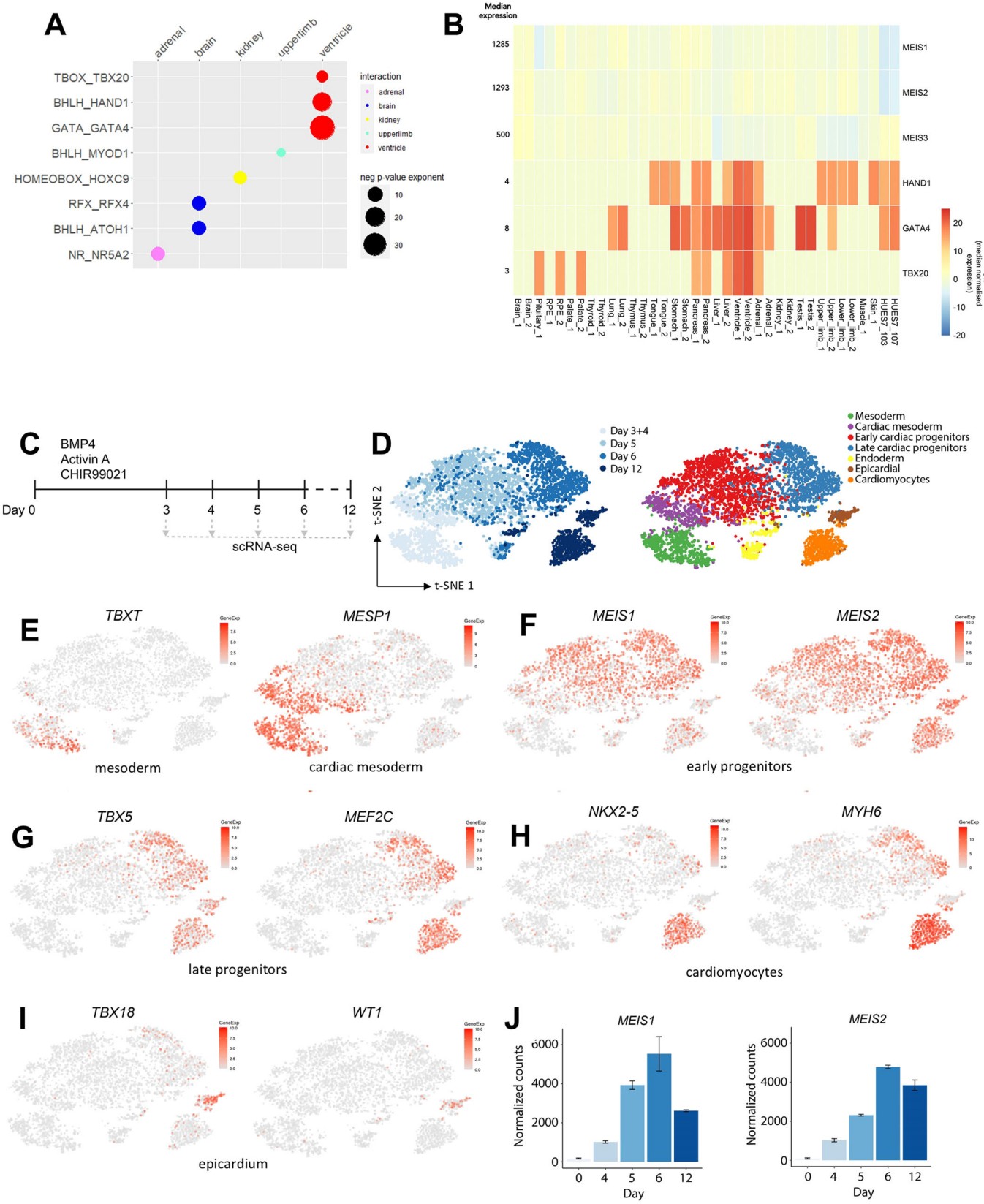

Figure 1.   Tissue-specific patterns of ubiquitous TFs.

(A) TALE binding site co-occurs with motifs recognized by tissue-restricted TFs at enhancers active in human embryonic development data sourced from (Gerrard et al, 2016). The tissue-restricted TFs listed on the left are predicted to bind signature motifs enriched in human enhancers specifically active in the tissue shown (top). TALE motif was identified as enriched within 100 nt upstream or downstream of the tissue-restricted TF motif. A cutoff of $P < 1 \times 10^{-5}$, calculated by HOMER using a cumulative binomial distribution, and enrichment in >5% of foreground regions was used. The size of the bubbles matches the negative log $P$ value: larger bubbles represented smaller $P$ values. (B) Expression of *MEIS1-2* and their predicted cardiac partners in human embryonic tissues; data sourced from (Gerrard et al, 2016). All tissues are duplicated except for the pituitary, muscle, and skin. Human embryonic RNA-seq read raw counts were down sampled based on the 75th percentile and read counts for each gene were divided by the gene's median read counts. Negative logarithms of the median normalized counts were used to plot the heatmap. (C) Schematic of cardiac differentiation protocol of hESCs and collection timepoints for scRNA-seq. (D) t-SNE plots of scRNA-seq cardiac differentiation time course colored by day and cell type. (E–I) t-SNE plots showing gene expression level of key markers for mesoderm (E), early (F) and late (G) progenitors, cardiomyocytes (H) and epicardial cells (I). Relative to the progression of differentiation, the earliest *MEIS1-* and *MEIS2-* positive cells correspond to early progenitors. (J) Bulk RNA-seq normalized counts for MEIS1 and MEIS2 of hESCs undergoing cardiac differentiation. n biological replicates = 2 (d0, d4, d5, d12); or = 3 (d6); the error bars indicate the standard deviation (SD).

contained in distal intergenic and intronic regions, while at d12, promoters account for half of the regions that increase in accessibility in the presence of MEIS. In contrast, regions whose accessibility increases in the absence of MEIS, are located in intergenic and intronic regions at both timepoints (Fig. EV3A). This discrepancy likely reflects the failure of mutant cells to progress to a fully differentiated phenotype, characterized by enhancer decommissioning and consequent loss of accessibility at distal and intronic regions (Wu et al, 2023). To identify chromatin and transcriptional changes initiated by MEIS, we mapped MEIS1 genomic occupancy in cardiac progenitors (d5), shortly after *MEIS1* expression. Most of our identified MEIS1 peaks were replicated in d6 cardiac progenitors (Data ref: Gonzalez-Teran et al, 2022a; Gonzalez-Teran et al, 2022b) (Fig. EV3B). High-confidence MEIS1 peaks (FE > 10; $n = 1742$; Dataset EV2) largely occur distally from any known transcriptional start site (TSS), with only a small fraction of MEIS binding (<5%) in the vicinity (within 5 kb) of a TSS (Fig. EV3C). GREAT (McLean et al, 2010) largely associated MEIS1 top peaks to genes involved in heart development (Fig. 3C). Chromatin accessibility and acetylation are hallmark features of active chromatin (Creyghton et al, 2010; Klemm et al, 2019). The top fraction of MEIS1 peaks almost exclusively intersect regions exhibiting increased accessibility in wild-type cells (Fig. EV3D). At the onset of differentiation (d5), we observed a significant reduction in chromatin accessibility (Figs. 3D and EV3E) and acetylation (Figs. 3E and EV3F) in MEIS KO cells; this shift is specific to regions normally occupied by MEIS1. The *PRDM6* and *TBX5* loci exemplify loss of chromatin accessibility and acetylation in the absence of MEIS (Fig. 3F). Finally, we assigned MEIS1 peaks to genes and intersected genes associated with top MEIS1 binding with DE genes at the same time point (d5). MEIS directly activates 133 genes (out of a total of 335 downregulated in the mutant) and repress 20 genes (out of a total of 122 upregulated in the mutant) (Fig. 3G). In sum, these results indicate that MEIS TFs predominantly function as transcriptional activators, promoting chromatin accessibility and acetylation at critical regions for cardiac differentiation, and the expression of their associated genes. The enrichment of intrinsically disordered regions (IDRs) in MEIS1 and MEIS2 also supports their role in activating transcription (Boija et al, 2018; Sabari et al, 2018) (Fig. EV3GH).

## MEIS cooperates with cardiac TFs

MEIS TFs are expressed in many non-cardiac cell types, yet they initiate a cardiac-specific transcriptional program. How do broadly expressed TFs execute lineage-specific functions? MEIS recognition motifs are enriched in tissue-specific enhancers, adjacent to motifs recognized by tissue-specific TFs (Fig. 1A). We hypothesized that MEIS could achieve tissue-specific functions through collaboration with lineage-specific partner TFs. To explore MEIS combinatorial binding in cardiac progenitors, we analyzed MEIS1 peaks using de novo motif discovery software (Heinz et al, 2010). We found MEIS recognition site as the second most enriched motif in MEIS1 peaks, with the top-ranked motif corresponding to HAND2 motif. GATA, HOX-PBX, and ZIC3 were also included in the top five over-represented motifs (Fig. 4A). Regions that lose accessibility in the absence of MEIS are enriched in largely similar motifs to those found in high-confidence MEIS peaks in cardiac progenitors (Fig. EV4A). Remarkably, regions that gain accessibility when transitioning from mesoderm to the cardiac progenitor state (Data ref: Bertero et al, 2019a; Bertero et al, 2019b), are enriched in the same motifs (Fig. EV4B), indicating that MEIS TFs operate as a hub in the network active in cardiac progenitors, which is fairly simple and dominated by few TFs at this stage. Next, we inspected up- and downregulated genes at d5 and d7, to identify TFs whose expression is directly regulated by MEIS. *HOXB1, HOXB2, GATA4* and *GATA6*, whose recognition sites are enriched in MEIS1 ChIP-seq and in regions that gain accessibility in cardiac progenitors, are also activated by MEIS1-2 (Fig. EV4C), with HAND TFs being a notable exception. Thus, MEIS positively control the expression of cardiac network TFs and cooperate with them.

To explore MEIS connectivity, we initially focused on GATA TFs. Given the established role of GATA4-5-6 in cardiac development (Tremblay et al, 2018), collaboration with these TFs could potentially explain how MEIS operate specifically at cardiac enhancers. Supporting MEIS-GATA combinatorial binding, GATA recognition motif is enriched in MEIS peaks (Fig. 4A), and MEIS and GATA sites co-occur in cardiac developmental enhancers (Fig. 1A). GATA motif is recognized by all six members of the GATA family (Tremblay et al, 2018), with *GATA4, GATA5,* and *GATA6* more highly expressed in d5 cardiac progenitors (Fig. EV4D). Experiments in mice and zebrafish suggest that the development of the cardiac system depends primarily on the overall dosage threshold of GATA proteins rather than on a specific GATA protein (Sam et al, 2020; Xin et al, 2006). To investigate MEIS and GATA connectivity, we mapped GATA6 binding in d5 wild-type and MEIS KO cardiac progenitors. We found that GATA6 binding partially overlaps with MEIS binding (Fig. 4B). In addition, GATA6 binding signal is significantly higher in the presence of MEIS1-2 (Fig. 4C). This increase is specifically observed when GATA6 occupies the same chromatin regions as MEIS1 and is therefore unlikely to be explained solely by the reduction in *GATA6* levels

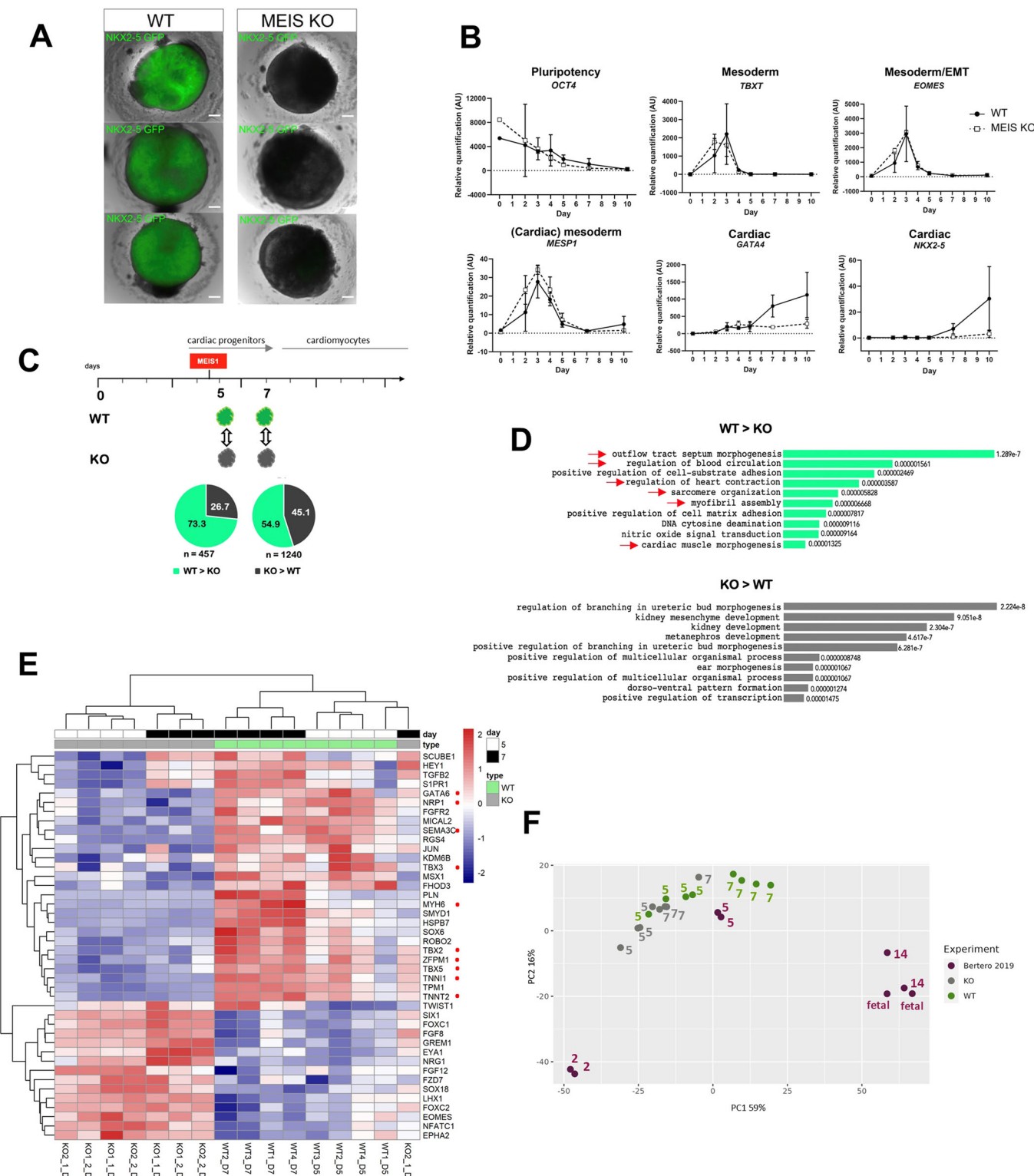

observed in MEIS KO. In contrast, the widespread spatial distribution of GATA6 is largely unchanged in the absence of MEIS (Fig. EV4E, see also Figs. 4D and EV4F). GATA6 extensive chromatin occupancy may reflect its early expression and regulatory roles in earlier differentiation stages or alternative cell fates. High-confidence MEIS1 and GATA6 peaks (FE > 10) are

associated with early cardiac phenotypes (Fig. EV4G) and are strategically positioned at critical genes for cardiac development and differentiation, including *TBX5, TNNT2, NRP1,* and also *HOXB1* (Figs. 4D and EV4F). Enhanced TF-binding levels and chromatin accessibility are hallmarks of TF cooperativity because TFs that bind together reinforce each other's occupancy on

◄ **Figure 2. MEIS1-2 are essential for cardiac differentiation.**

(A) Live imaging of cardiac lineage reporter NKX2-5-GFP in wild-type and MEIS KO EBs at d12. In the same conditions, MEIS KO EBs fail to activate NKX2-5 expression. The same phenotype was consistently observed in three MEIS KO clones (see also Fig. EV2B). The GFP channel is overlayed on the brightfield images. Scale bar = 200 μm. (B) Relative gene expression of pluripotency, mesoderm, and cardiac mesoderm markers, measured by qPCR, in wild-type and MEIS KO hESC undergoing cardiac differentiation. AU arbitrary units. N = 2 biological replicates; the error bars indicate the SD. (C) Schematic of the comparative analysis of wild-type and MEIS KO transcriptomes with the proportion of down- and upregulated genes at d5 and d7 of cardiac differentiation. The onset of MEIS1 activation is indicated above the timeline. (D) GO associated with genes down- and upregulated in the absence of MEIS1-2. Genes downregulated in the MEIS KO at d5 (green) are almost exclusively associated with cardiovascular terms while upregulated genes (dark gray) display a variety of developmental GO terms, none associated with cardiac development. Only the top 10 GO terms are shown. (E) Expression of cardiac mesoderm GO genes in wild-type (green) and MEIS KO (gray) cells at d5 (white) and d7 (black). Only genes with significant changes in expression at d5 (log FC > [1]; Padj <0.05) are plotted. Red dots indicate genes with a cardiac phenotype or cardiac-specific expression. (F) PCA plots comparing human cardiac differentiation (d2, d5, d14) and fetal heart (fetal) transcriptomes (purple), sourced from (Bertero et al, 2019b) alongside d5 and d7 wild-type (green) and MEIS KO (gray) transcriptomes. D5-7 MEIS KO cells cluster more closely than wild-type cells along PC1 (representing time), indicating a disruption in the trajectory toward cardiac differentiation. Source data are available online for this figure.

chromatin, outcompeting nucleosomes for DNA access. In addition to MEIS1 increasing GATA6 chromatin occupancy, sites co-occupied by MEIS1 and GATA6 exhibit significantly higher accessibility (Figs. 4E and EV4H) and acetylation (Fig. EV4I) compared to sites occupied by MEIS1 alone or GATA6 alone. These findings collectively support cooperative binding interactions between MEIS and GATA. To address if MEIS-GATA combined binding has a functional impact, we analyzed data from a GATA4 loss-of-function model in iPSC-derived cardiac progenitors (Gonzalez-Teran et al, 2022b). Notably, we found enrichment of MEIS motifs in GATA4 peaks within this system (Fig. EV4J). Consistent with functional synergy, 21% of MEIS direct target genes (downregulated in MEIS KO) exhibit downregulation in GATA4 knockdown within iPSC-derived cardiac progenitors (Data ref: Gonzalez-Teran et al, 2022a; Gonzalez-Teran et al, 2022b) (Fig. 4F). Of note, the observation that GATA6 binds to genes co-regulated by MEIS and GATA4, even in the absence of MEIS (Figs. 4D and EV4F), suggests that GATA occupancy alone is not sufficient; MEIS is required for full gene activation. The enrichment of GATA motifs in MEIS peaks, and vice versa, implies a model wherein MEIS and GATA recruitment to shared enhancers is DNA-guided (Fig. 4G). In addition, when overexpressed, both MEIS1 and MEIS2 can immunoprecipitate GATA4 and GATA6, indicating MEIS and GATA TFs can also form a complex (Fig. 4H). In summary, the presence of nearby recognition motifs and protein-protein interactions underlie MEIS and GATA combinatorial binding, resulting in increased chromatin accessibility and the coordinated regulation of genes crucial for cardiac development and differentiation.

## Dynamic competition in MEIS combinatorial binding: balancing HOX and GATA interactions

MEIS peaks are also enriched in motifs recognized by HOX TFs, and HOXB1 and HOXB2 are significantly downregulated in MEIS KO (Figs. 4A and EV4C). The interaction between HOX and MEIS TFs is well-established in a variety of organisms and tissues (Bobola and Sagerström, 2024), but its significance in cardiac development remains underexplored. The human HOX family includes 39 members, which recognize highly similar sequence motifs (Rezsohazy et al, 2015). We first asked which of the HOX genes are expressed in d5 cardiac progenitors. We found that anterior members of the HOXB cluster exhibit the highest expression levels (Fig. EV5A), including HOXB1 and HOXB2 that are significantly downregulated in the MEIS KO (Fig. EV4C). In mouse, overexpression of HOXB1 in the second heart field (SHF) results in

arrested cardiac differentiation; the effects of the overexpression are entirely consistent with the phenotype of Hoxa1[-/-]; Hoxb1[-/-] loss-of-function hearts (Stefanovic et al, 2020). To investigate the role of HOX proteins in this system, we chose a similar overexpression approach to circumvent the issue of redundancy often encountered in HOX mutants. We generated lines where HOXB1 is placed under a doxycycline-inducible promoter at a safe harbor locus (Zhu et al, 2014). HOXB1 expression was induced just before the onset of MEIS1 expression and maintained until d9 of cardiac differentiation (Fig. 5A). At d9, elevated HOXB1 dosage led to a reduction in GFP-positive, NKX2-5 expressing cells, which exhibited little to no contraction compared to their control counterparts, indicating a delay in cardiac differentiation. In addition to NKX2-5, the downregulation of key genes like TNNT2 and MYH6, encoding for cardiac-specific troponin and myosin, which function in cardiac muscle contraction, further underscores HOXB1 inhibitory effect on cardiac muscle differentiation (Fig. 5B). In sum, overexpression of HOXB1 has a negative effect on cardiac differentiation, which is consistent with Hoxb1 overexpression in vivo and in cardiac differentiation of mouse ESC (Stefanovic et al, 2020). Unsurprisingly, both MEIS1 and MEIS2 demonstrate the capacity to interact with HOXB1, given their well-established partnership with HOX proteins (Fig. 5C). Cardiomyocyte differentiation involves several regulatory waves that lead to activation of progenitor-specific genes followed by activation of cardiac-specific genes. We examined HOXB1 and GATA expression in a time course of cardiac differentiation. We found distinct expression dynamics: HOXB1 expression peaks during the progenitor stage (d5–d7) and then declines to basal levels in cardiomyocytes. Conversely, expression of GATA4 increases with cardiomyocyte differentiation, and both GATA4 and GATA6 remain highly expressed in cardiomyocytes (Fig. 5D). These contrasting expression dynamics suggest that HOX and GATA may act independently to control distinct aspects of cardiac lineage differentiation, rather than acting in concert. Consistent with this hypothesis, HOXB1 and GATA exhibit seemingly opposite effects on cardiac differentiation: while HOXB1 delays cardiac differentiation, GATA4-5-6 are known for their established cardiogenic effects, with GATA4 being one of three TFs capable of converting fibroblasts into cardiomyocytes (Ieda et al, 2010). These observations, together with the finding that GATA4 and HOXB1 interact with MEIS1 (Figs. 4H and 5C), suggest a potential competition for MEIS binding between these two TF families. Supporting this hypothesis, the addition of HOXB1 reduced the level of GATA4 co-precipitated by MEIS1 to 40% (Fig. 5E). Similarly, the addition of GATA4 decreased the amount

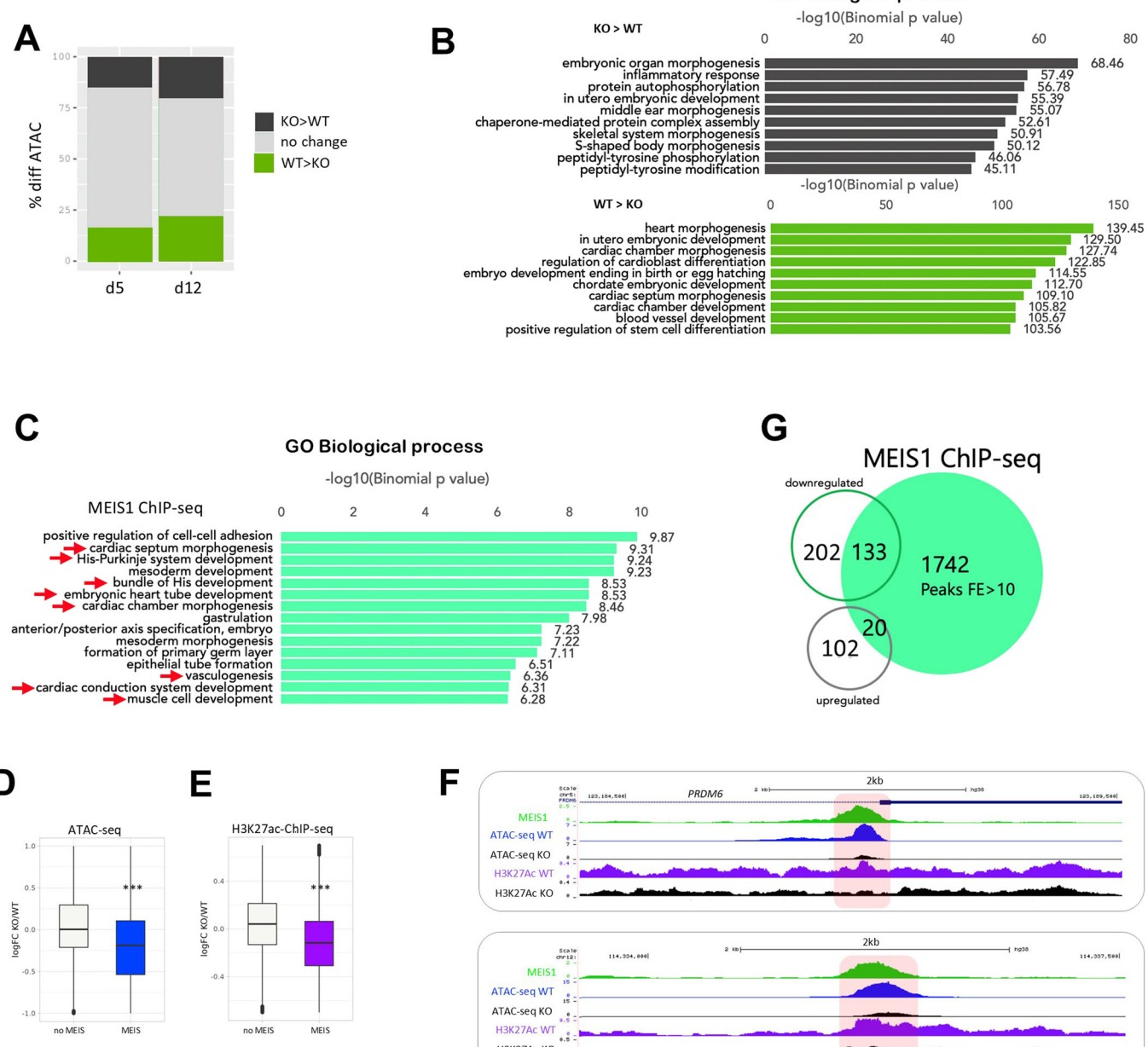

of HOXB1 co-precipitated by MEIS1. Co-precipitation assays using MEIS1 domains revealed that both HOXB1 and GATA4 primarily interact with the N-terminal, including the homeodomain (HD), of MEIS1. Notably, GATA4 interacts almost exclusively with this region, showing minimal binding to the C-terminal of MEIS1 (Fig. EV5B). To test if HOX-GATA competitive binding affects MEIS genomic occupancy, we altered HOX-GATA balance during cardiac differentiation. We used our HOXB1 overexpression model to maintain high HOXB1 dosage up to d8, when HOXB1 levels are declining, and GATA4 expression is steadily increasing. We first selected HOXB1-MEIS target enhancers based on specific criteria, including overlap with a top MEIS1 peak, increased accessibility in wild-type cells, presence of at least one HOX-PBX consensus motif, and association with a DE gene in MEIS KO cells. Subsequently, we validated HOXB1 binding to these regions using ChIP-seq (Fig. 5F).

Next, we identified GATA-MEIS target enhancers, defined as regions bound by GATA4 and GATA6 (Fig. 5F), overlapping with MEIS1 top peaks and linked to genes downregulated in GATA4 knockdown cardiac progenitors (Data ref: Gonzalez-Teran et al, 2022a; Gonzalez-Teran et al, 2022b). We found that sustained high dosage of HOXB1 in d8 cardiomyocytes leads to increased MEIS binding levels at HOXB1-bound enhancers (Figs. 5G and EV5C). The expression of genes associated with these enhancers is also increased, suggesting a functional effect (Fig. 5H). Conversely, in the same HOXB1-overexpressing cells, we observed a reduction in MEIS1-binding levels at high-confidence GATA enhancers, correlating with decreased expression of genes associated with these enhancers (Figs. 5GH and EV5C). These results suggest that HOXB1 may delay cardiac differentiation by diverting MEIS away from GATA-bound, cardiac-specific enhancers. Supporting this

**Figure 3. MEIS TFs promote chromatin accessibility and enhancer activation at critical regions for cardiac differentiation.**

(A) Bar chart of chromatin accessibility shows unchanged accessibility at most chromatin regions in the absence of MEIS (d5 and d12). (B) GREAT analysis of differential ATAC peaks (d5). Regions more accessible in wild-type (WT > KO, green bars) are associated to cardiac biological processes. In contrast, ATAC-seq peaks higher in mutant (KO > WT, dark gray bars) shows association with genes involved in a variety of biological processes, including non-cardiac organ development and inflammation. The length of the bars corresponds to the binomial raw (uncorrected) $P$ values ($x$ axis values). (C) GREAT analysis of MEIS1 high-confidence peaks (FE > 10) at d5. MEIS1 peaks predominantly cluster around genes specifically associated with cardiac biological processes (red arrows). (D, E) Global changes in accessibility (D) and H3K27ac (E) in wild-type versus MEIS KO cells at d5 of cardiac differentiation, expressed as log2 FC of normalized count values. The absence of MEIS TFs causes a significant reduction in accessibility and H3K27ac at MEIS1-occupied regions, identified by Motif2Sites, relative to unbound regions (one-sided $t$ test; ***$P < 2.2e^{-16}$). The box plots represent the distribution of (D) ATAC-seq log FC values within the range of $-1$ (minima) to 1 (maxima), and (E) H3K27ac log FC values within the range of $-0.5$ (minima) to 0.5 (maxima). The samples include 4 biological replicates for ATAC-seq (2 wild-type and 2 MEIS KO) and 4 biological replicates for H3K27ac ChIP-seq (2 wild-type and 2 MEIS KO). The center (line inside the box) marks the median value. The box bounds span the interquartile range (IQR), from the first quartile (Q1) to the third quartile (Q3). The whiskers extend to 1.5 times the IQR beyond Q1 and Q3. Data points outside this range are plotted individually. (F) Loss of accessibility and acetylation at MEIS-regulated genes. UCSC tracks of MEIS1 binding (green), ATAC-seq in wild-type (blue) and MEIS KO (black), and H3K27Ac ChIP-seq in wild-type (purple) and MEIS KO (black) at representative loci, *PRDM6* (top) and *TBX5* (*TBX5* transcription termination site is located 16 kb downstream). Differential peaks are shaded in pink. (G) MEIS1 predominantly function as an activator. Intersection of genes down- and upregulated at d5 with genes associated to d5 MEIS1 high-confidence (FE > 10) peaks (using GREAT rules). The majority of genes dysregulated in MEIS KO and exhibiting MEIS1 binding nearby are activated by MEIS TFs (downregulated in MEIS KO).

interpretation, most HOX-MEIS binding is excluded from GATA-MEIS regions and is associated with the regulation of various processes, not just those related to cardiac differentiation, as observed for GATA-MEIS regions (Fig. EV5D). In sum, MEIS acts as a molecular switch, trading partners to control the progression of cardiac differentiation. A similar separation of GATA6-MEIS and HOXA3-MEIS binding, each associated with distinct biological processes, is observed in the cardiac neural crest-populated posterior branchial arches (PBA) (Data ref: Losa et al, 2017a; Losa et al, 2017b); (Data ref: Bridoux et al, 2020b; Bridoux et al, 2020a); (Data ref: Bridoux et al, 2020c; Bridoux et al, 2020a) (Fig. EV5D). This indicates that HOX and GATA diversify MEIS's regulatory functions and suggests that their mutual competition for MEIS is a general mechanism.

## MEIS promotes the accumulation of KMT2D at cardiac enhancers

While it is well-established that lineage-specific TFs are essential for lineage-specific transcription, our results indicate that broadly expressed TFs are also essential. What function do they provide in cell differentiation programs? Based on the association of MEIS1 binding with transcriptional activation, a likely hypothesis is that MEIS TFs harness a general function, critical for the activation of gene expression. A clue to the nature of this general function comes from the observation that MEIS2 syndrome, caused by mutations in the *MEIS2* gene, display clinical features – distinctive facial appearance, developmental delay, intellectual disability, heart and skeletal abnormalities - that closely resembles those of Kabuki syndrome (Gangfuß et al, 2021). Kabuki syndrome is caused by mutation in the *KMT2D* gene (Ng et al, 2010), encoding for a ubiquitously expressed histone-lysine N-methyltransferase, which monomethylates lysine 4 of histone H3 (H3K4me1) of distal enhancers, leading to transcriptional activation (Hu et al, 2013). KMT2D lacks a DNA binding domain and relies on DNA binding proteins to identify its correct genomic addresses (Froimchuk et al, 2017). Similar to MEIS, KMT2D is required for cardiac development (Ang et al, 2016). We surveyed available datasets for evidence of an interaction between MEIS1 and KMT2D. Dysregulation of KMT2 proteins (MLL) and MEIS1 contribute to acute myeloid leukemia (AML). A physical association between MEIS1 and KMT2D (MLL4) has been reported in AML cell lines (Aubrey et al, 2022), where KMT2D is identified among the top 50 MEIS1

interactors. Next, we asked if MEIS1 and KMT2D bind common regions. Top KMT2D peaks associated with high-confidence MEIS1-bound regions in cardiac progenitors are found nearby genes involved in cardiac development and differentiation, in contrast to non-overlapping KMT2D peaks, mostly associated with metabolic pathways (Fig. 6A). We confirmed MEIS1 chromatin colocalization with KMT2D using reChIP (Fig. 6B). These results indicate that MEIS1 and KMT2D bind together to a subset of regions. To establish causality, we interrogated KMT2D binding in the absence of MEIS1-2. We found that KMT2D levels are significantly higher in wild-type cells relative to MEIS KO cells, and this effect is specific to regions bound by MEIS1 (Fig. 6C,D). This trend is exemplified by the *ZNF503* locus, where KMT2D signal is specifically decreased at regions overlapping MEIS1 peak. In contrast KMT2D binding at regions not overlapping MEIS1 peaks appears unchanged (Fig. 6E). KMT2D binding sites that show decreased binding levels in the absence of MEIS1-2, are predominantly co-occupied by MEIS1 in combination with GATA4-6 (Fig. 6F). Conversely, only a small proportion of regions with increased KMT2D binding in MEIS KO cells, overlap with MEIS or GATA binding sites (Fig. 6G). Together, these observations suggest that MEIS1-2 function to recruit KMT2D at lineage-specific, GATA-occupied enhancers to activate the cardiac gene expression program. KMT2D monomethylates histone H3 at enhancers active during cell fate transitions, eventually promoting the recruitment of p300 and deposition of H3K27Ac (Lai et al, 2017; Lee et al, 2013). We explored the distribution of H3K27Ac signals in wild-type cells relative to MEIS KO. H327Ac signal is significantly higher at regions where both MEIS1 and KMT2D are present indicating a synergy between MEIS1 and KMT2D to enhance chromatin acetylation (Fig. 6H). In summary, MEIS proteins function as *actuator* TFs in cardiac cells. They initiate cardiac-specific transcription by harnessing general co-activators at cardiac-specific enhancers, to which they directed by binding with lineage- and/or domain-specific TFs (Fig. 7).

## Discussion

Combinatorial binding is a common feature of mammalian transcription networks, where TFs collaborate to access DNA and enhance their chromatin binding. This process allows for binding and response specificity, broadening the regulatory capabilities of

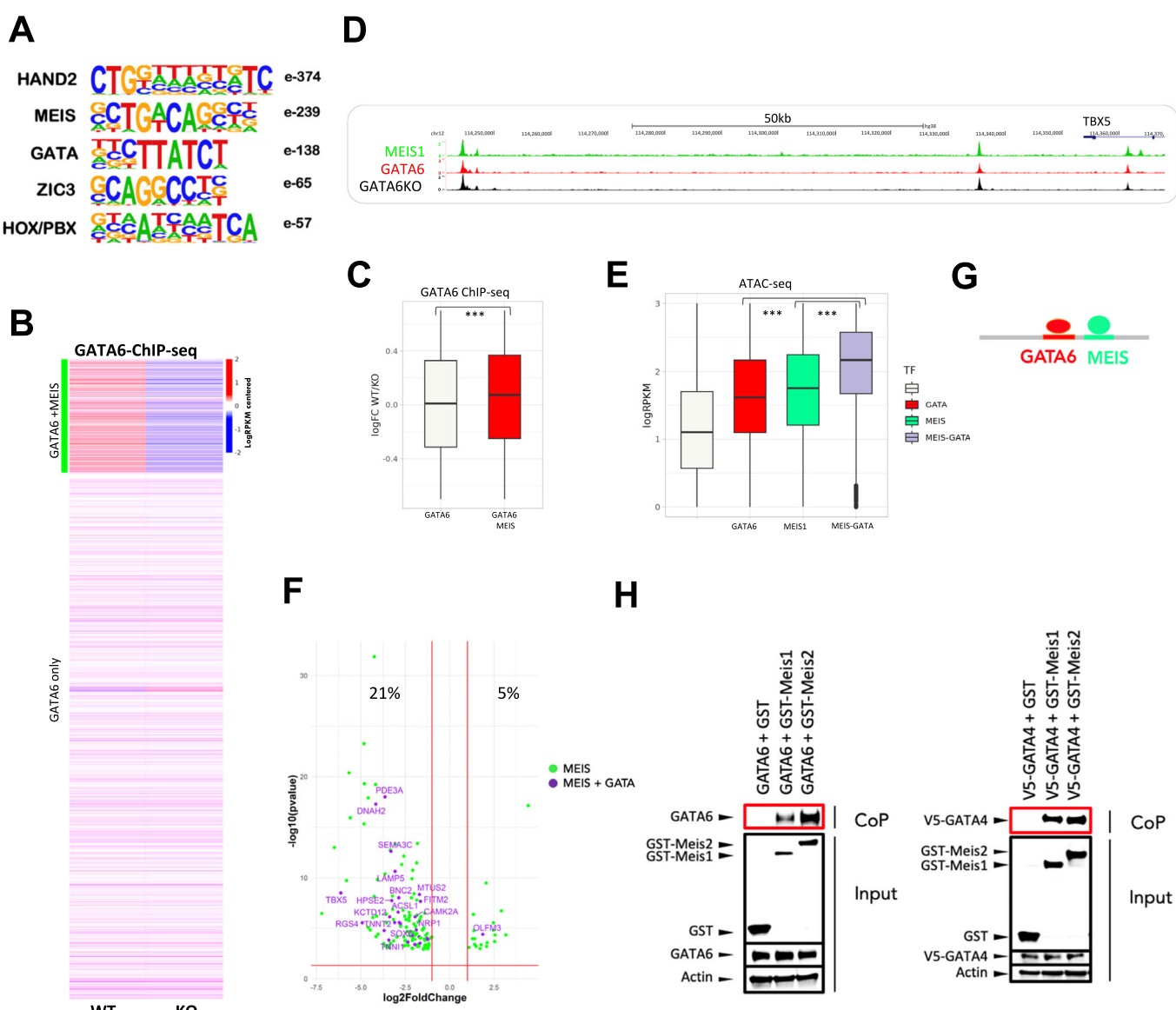

**Figure 4. MEIS direct cooperativity with GATA TFs.**

(A) Top motifs, ranked by significance, identified in MEIS1 peaks using de novo motif discovery (Homer). (B) GATA6 binding signal in d5 progenitors. Average logRPKM values of GATA6 replicate experiments, mean-centered across wild-type and MEIS KO samples. The total number of GATA6 binding sites identified by Motif2Sites is 198,879, of which 35229 are occupied by MEIS1. (C) Quantitative changes in GATA6 binding levels at MEIS-bound location. A boxplot of GATA6 differential binding (log FC wild-type vs. MEIS KO) shows that MEIS proteins significantly increase GATA6 binding at MEIS1-occupied sites. In contrast, GATA6 binding at MEIS1-unbound regions is unaffected by MEIS proteins (one-sided $t$ test; ***$P < 2.2e^{-16}$). The box plots represent the distribution of GATA6 log FC values within the range of $-0.5$ (minima) to 0.5 (maxima). The samples include 4 biological replicates for GATA6 ChIP-seq (2 wild-type and 2 MEIS KO). The center (line inside the box) marks the median value. The box bounds span the interquartile range (IQR), from the first quartile (Q1) to the third quartile (Q3). The whiskers extend to 1.5 times the IQR beyond Q1 and Q3. Data points outside this range are plotted individually. (D) UCSC tracks of MEIS1 (green) and GATA6 in wild-type and MEIS KO d5 progenitors (red and black, respectively) at the *TBX5* locus. GATA6 peaks are still detected in the absence of MEIS1-2 (see also Fig. EV3). (E) Co-binding of GATA6 and MEIS1 increases chromatin accessibility. A boxplot of accessible chromatin levels at GATA6 and MEIS1-occupied regions. Globally, regions co-occupied by GATA6 and MEIS1 have significantly higher accessibility relative to regions occupied by either MEIS1 or GATA6 (one-sided $t$ test; ***$P < 2.2e^{-16}$). The box plots represent the distribution of logRPKM values within the range of 0 (minima) to 3 (maxima). The samples include two biological replicates for ATAC-seq (wild-type). Center, box bounds and whiskers are defined as in (C). (F) Genes co-regulated by MEIS and GATA4 TFs in cardiac progenitors. Volcano plot of DE genes in MEIS KO that are associated with a MEIS1 peak in d5 cardiac progenitors. Purple dots indicate genes that are also downregulated in GATA4 knockdown d6 iPSCs-derived cardiac progenitors; data sourced from (Gonzalez-Teran et al, 2022b). Green dots indicate MEIS-only targets. FC and $P$ values for DE genes were calculated using DESeq2, based on 8 RNA-seq experiments (4 wild-type and 4 MEIS KO biological replicates). (G) Model of indirect cooperativity: co-binding of GATA and MEIS TFs is guided by the vicinity of their recognition motifs. (H) Co-precipitation assays. HEK293 cells were co-transfected with expression vectors for GATA6 or V5-tagged GATA4 and GST-tagged MEIS1 or MEIS2 or GST alone. Protein interactions were assayed by co-precipitation on glutathione beads directed toward the GST tag, and eluted proteins were analyzed by western blotting to detect the presence of GATA6 and V5-GATA4 (red box, CoP). Cell lysates were analyzed by western blotting prior to co-precipitation to detect protein expression (input), including ubiquitously expressed actin, used as a control. Source data are available online for this figure.

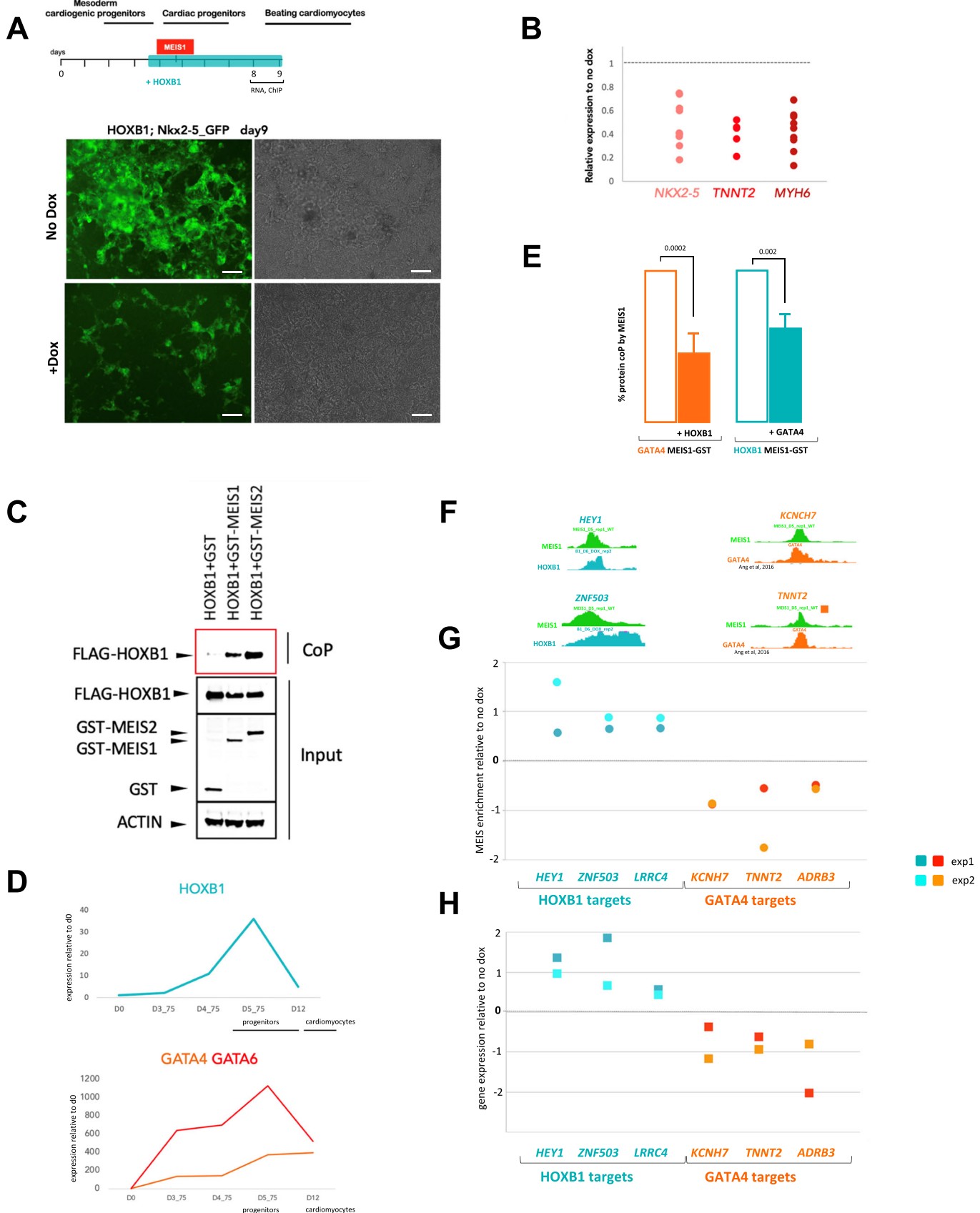

**Figure 5.   MEIS combinatorial binding drives the dynamics of cardiac lineage differentiation.**

(A) Schematic of *HOXB1* overexpression along the timeline of cardiac differentiation. At d9, HOXB1; NKX2-5-GFP cells display strong GFP expression, mirroring NKX2-5 expression. HOXB1 induction with doxycycline results in fewer cells expressing GFP. Scale bar = 150 μm. (B) Cardiomyocyte markers are downregulated in d9 HOXB1-overexpressing cells. *NKX2-5, MYH6* and *TNNT2* expression, measured by qPCR, is expressed as a percentage relative to their expression measured in control, non-induced cells. (C) Co-precipitation assays. HEK293 cells were co-transfected with expression vectors for FLAG-tagged HOXB1 and GST-tagged MEIS1 or MEIS2, or GST alone. Protein interactions were assayed by co-precipitation on glutathione beads directed toward the GST tag and eluted proteins analyzed by western blotting to detect the presence of FLAG-HOXB1 (red box, CoP). Cell lysates were analyzed by western blotting prior to co-precipitation to detect protein expression (input), including ubiquitously expressed actin, used as a control. (D) Dynamic expression of *HOXB1* and *GATA4-6* during cardiac differentiation. Time course of cardiac differentiation analyzed by RNA-seq. At each time point shown, normalized transcript counts are expressed as fold change relative to their value at the start of differentiation (d0). (E) HEK293 cells were co-transfected with expression vectors for GATA4, GST-tagged MEIS1, and FLAG-tagged HOXB1. Co-precipitation of GATA4 with MEIS1-GST in the presence of HOXB1-FLAG was detected by western blotting and is expressed as a relative percentage of GATA4 co-precipitated with MEIS1-GST in the absence of HOXB1 (orange bars; Tukey HSD test $P < 0.0002$; $n = 3$). Co-precipitation of HOXB1 with MEIS1-GST in the presence of GATA4 was detected by western blotting and is expressed as a relative percentage of HOXB1 co-precipitated with MEIS1-GST in the absence of GATA4 (teal bars; Tukey HSD test $P < 0.002$; $n = 3$ biological replicates). The error bars indicate the SD. (F) UCSC tracks of MEIS1 (green) in d5 cardiac progenitors, HOXB1-FLAG (teal) in d8 cardiomyocytes and GATA4 (orange) in d6 cardiac progenitors at representative HOXB1- MEIS1 target regions (left) and GATA-MEIS regions (right). (G, H) MEIS1-binding levels (G) and gene expression (H) in d8 cardiomyocytes with sustained HOXB1 dosage. (G) ChIP qPCR using MEIS1 antibody in d8 dox-induced (HOXB1-overexpressing) and -dox (control) differentiating cardiomyocytes. Scatter plot of MEIS1-binding levels measured at HOX-MEIS target regions (blue) and GATA-MEIS-bound regions (orange). MEIS1 enrichment (relative to input) is plotted as log FC relative to MEIS1 enrichment measured at the same region in the control. Blue dots represent log FC value of MEIS1 enrichment measured on HOXB1-target enhancers in two independent experiments. Orange and red dots represent log FC value of MEIS1, measured on GATA4-target enhancers in two independent experiments; see also Fig. EV5B). (H) Expression of *HEY1, ZNF503, LRRC4* (MEIS1-HOXB1 targets) and *KNCH7, TNNT2, ADRB3* (MEIS1-GATA4 targets), measured by qPCR, and expressed as log FC of their relative expression measured in control, non-induced cells. Source data are available online for this figure.

the finite set of TFs encoded by an organism. However, it is less clear if combinatorial binding also functions to assemble complexes with emerging functional properties in transcriptional activation. Here, we identify the essential role of the broadly expressed, non-lineage-specific TFs MEIS in activating lineage-specific transcription and establishing cardiac fate. We propose that MEIS proteins are obligatory components of productive TF combinatorial binding, functioning as actuator TFs to effectively activate the transcriptional programs selected by lineage- or domain-enriched TFs (GATA and HOX). MEIS TFs, though largely ubiquitous, are selectively guided to cardiac-specific enhancers by binding with lineage-enriched TFs. Increasing the binding levels of actuator TFs at cardiac enhancers activates gene expression. This happens, at least in part, because MEIS TFs promote the accumulation of the methyltransferase KMT2D to initiate enhancer commissioning.

MEIS TFs significantly promote transcription, predominantly eliciting positive effects. This aligns with previous observations of high MEIS binding correlation with enhancer activity and gene activation across different tissues of the developing mouse embryo (Bridoux et al, 2020a). TALE TFs are evolutionarily ancient and were present in unicellular eukaryotes before the diversification of metazoans (Joo et al, 2018; Mukherjee and Bürglin, 2007). An intriguing possibility is that, with the advent of multicellularity and the generation of specialized cell types, new tissue-restricted proteins (e.g., HOX) evolved within the genetic toolkit of animals to harness the function of these ancient activators in a cell type- or tissue-specific manner.

This new framework suggests a strategy for efficient cell reprogramming. Lineage-specific TFs are the primary candidates for stem cell differentiation and reprogramming protocols, yet combinations of these TFs or those enriched in specific lineages often prove inefficient in generating the desired cell type (Wang et al, 2021). Combining lineage-specific TFs with broadly expressed TFs, which are often overlooked in tissue-specific processes, could significantly enhance the efficiency of direct reprogramming.

Congenital heart disease represents a significant birth defect, impacting up to 1% of all live births in the Western world (Hoffman and Kaplan, 2002). Extensive research into cardiac development and differentiation has identified numerous TFs and

signaling pathways as critical components in vertebrate heart formation (Cui et al, 2018). Disruption of *Meis1* in mouse, as well as of *Pbx* genes, causes cardiac outflow tract defects (Stankunas et al, 2008). Previous studies have suggested a potential role for MEIS proteins in regulating cardiomyocyte differentiation (Bouilloux et al, 2016; Dupays et al, 2015; Paige et al, 2012; Quaranta et al, 2018; Wamstad et al, 2012), as well as proliferation (Nguyen et al, 2020) and functionality of postnatal cardiomyocytes (Liu et al, 2022), but the requirement of MEIS TFs for human cardiac differentiation has remained largely unrecognized. Here, we identify MEIS TFs as essential regulators of human cardiomyocyte differentiation. MEIS function as a central hub, enabling and coordinating the progression of cardiac differentiation.

MEIS KO has a profound impact on cardiac differentiation, affecting both cardiac and epicardial lineages. Motif analysis indicates that MEIS co-binding partners are also involved in the transition from mesoderm to cardiac progenitors, highlighting MEIS as a key node of cardiac transcription networks. TFs cooperate through direct and indirect mechanisms. MEIS1-2 physically interact with HOXB1 and GATA4-6, pointing to a mechanism of direct protein-protein cooperativity, though these interactions have been observed in overexpression experiments, which may not reflect physiological conditions. In addition, GATA and HOX recognition motifs are enriched in MEIS peaks, and MEIS motifs are significantly enriched near GATA motifs in human developmental enhancers. This also supports a mechanism of indirect cooperativity, where TFs bind closely spaced sites on chromatin and rely on mutual interdependence to access their respective sites on chromatin. No evidence of combined binding sites was found, indicating that complex formation does not alter the binding capabilities of its individual components. Several lines of evidence suggest that MEIS can bind alternately, but not simultaneously, with HOX and GATA. Firstly, GATA and HOX can outcompete each other for binding to MEIS. Secondly, HOX and GATA exhibit opposite effects on cardiac differentiation. Thirdly, a dynamic temporal expression pattern is observed: HOX levels are high in cardiac progenitors and low in cardiomyocytes, while GATA4 levels show the opposite pattern. Variations in TF dosage regulate the distribution of MEIS at enhancers. Maintaining high levels of HOX beyond the progenitor stage, delays cardiac differentiation. At the

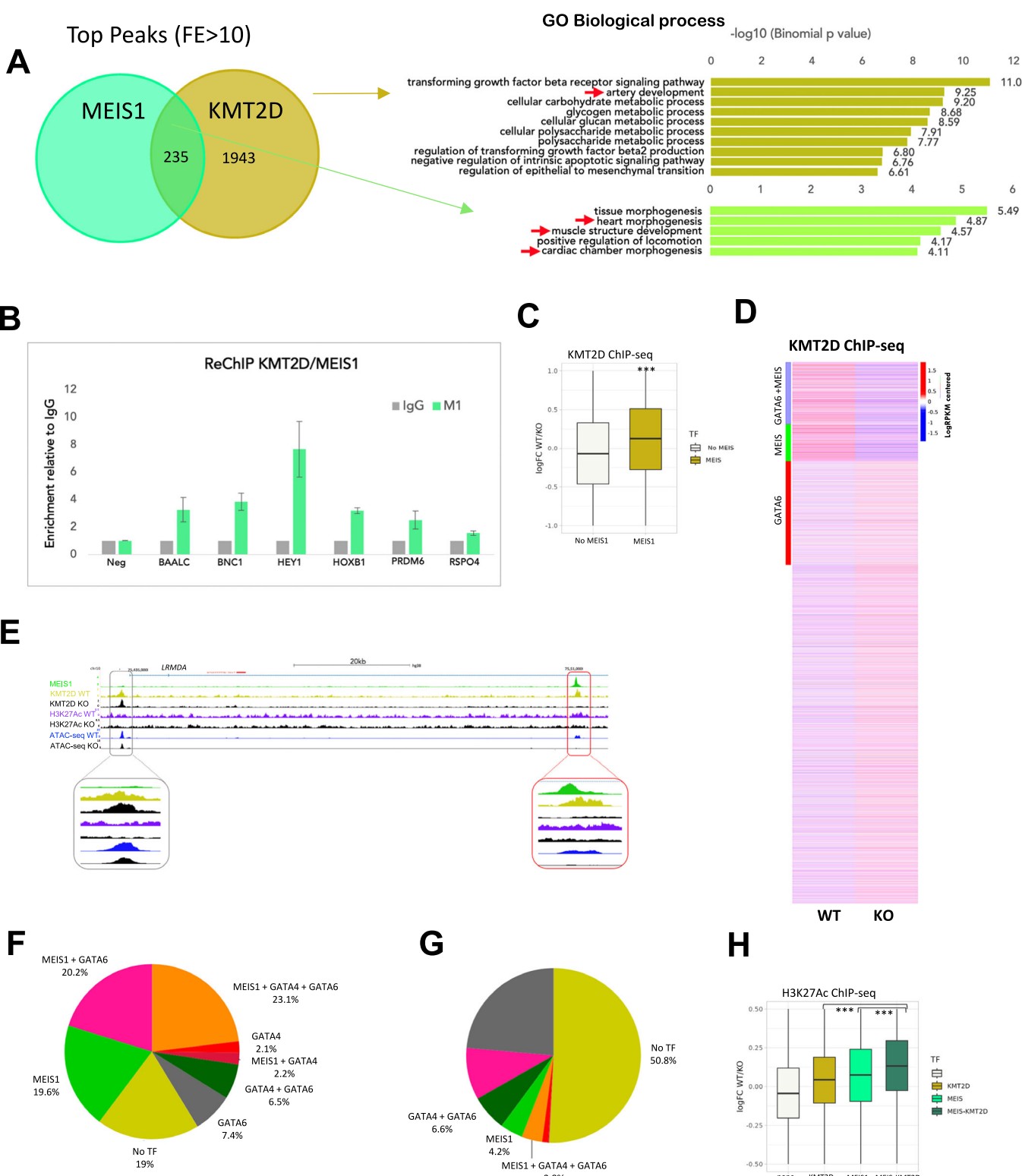

genomic level, MEIS is redistributed from GATA-bound enhancers to HOX-bound enhancers, leading to changes in the expression of the associated genes. HOX proteins are well-established partners of MEIS proteins (Bobola and Sagerström, 2024). They control embryonic patterning in all animals, deploying different mechanisms, including

control of cell fate (Rezsohazy et al, 2015). In mouse, HOXB1 maintains cardiac progenitors, as its depletion accelerates cardiac differentiation (Stefanovic et al, 2020). Given the central role of MEIS in cardiac differentiation, one possible mechanism for HOX action is to restrict the availability of MEIS, preventing it from interacting with

Figure 6.    MEIS promotes the accumulation of KMT2D at cardiac enhancers.

(A) Intersection of MEIS1 and KMT2D top peaks (FE > 10) in d5 cardiac progenitors. Shared MEIS1-KMT2D peaks are mainly associated with genes implicated in cardiac GO. In contrast, KMT2D only peaks cluster around genes involved in metabolic processes. The length of the bars corresponds to the binomial raw (uncorrected) *P* values calculated by GREAT using a Binomial Test. (B) Re-ChIP using KMT2D antibody, followed by MEIS1 antibody. KMT2D immunoprecipitated chromatin was again precipitated using MEIS1 or an unspecific (IgG) antibody and amplified using primers spanning KMT2D and MEIS1 co-occupied regions. KMT2D-occupied regions are enriched upon MEIS1 immunoprecipitation, while no enrichment is observed using a negative control region; the negative control is an unbound region. MEIS1-binding enrichment is expressed relative to IgG. $n = 2$ biological replicates; the error bars indicate the SD. (C) Boxplot of KMT2D log FC in wild-type versus MEIS KO cells. The presence of MEIS TFs significantly increases KMT2D signal at MEIS1-occupied regions. In contrast, no significant change is observed at regions that are not bound by MEIS1 (one-sided *t* test; ***$P < 2.2e^{-16}$). The box plots represent the distribution of KMT2D log FC values within the range of $-1$ (minima) to 1 (maxima). The samples include four biological replicates for KMT2D ChIP-seq (2 wild-type and 2 MEIS KO). The center (line inside the box) marks the median value. The box bounds the interquartile range (IQR), from the first quartile (Q1) to the third quartile (Q3). The whiskers extend to 1.5 times the IQR beyond Q1 and Q3. Data points outside this range are plotted individually. (D) KMT2D ChIP-seq average logRPKM values of replicates, mean-centered across d5 wild-type and MEIS KO samples. Short reads were counted in open chromatin regions from ATAC-seq ($= 142411$), grouped into GATA6 + MEIS1 (16315), MEIS1 only (9747) and GATA6 only (27242). (E) UCSC tracks of MEIS1 binding (green), ATAC-seq in wild-type (blue) and MEIS KO (black), H3K27Ac ChIP-seq in wild-type (purple) and MEIS KO (black) and KMT2D ChIP-seq in wild-type (gold) and MEIS KO (black) at the *ZNF503* locus. Differential KMT2D peaks, boxed in red, overlap MEIS1 peaks. In contrast, loss of MEIS1 does not affect KMT2D binding at MEIS1-unbound regions (boxed in gray). Enlarged views of the same regions are shown below the tracks. (F) Overlap of KMT2D differential binding (KMT2D WT > KO log FC > 1, total of 5498 regions) with MEIS1, GATA6 and GATA4. Percentages are rounded to one decimal. (G) Overlap of KMT2D differential binding (KMT2D KO > WT log FC > 1, total of 5114 regions) with MEIS1, GATA6 and GATA4. Percentages are rounded to one decimal. Only values above 1% are shown. (H) Boxplot of H327Ac log FC in wild-type versus MEIS KO cells. Relative to regions occupied by KMT2D or MEIS1 alone, regions co-occupied by MEIS1 and KMT2D display increased acetylation in wild-type versus MEIS KO cells, suggesting that co-occupancy of MEIS1 and KMT2D enhances chromatin acetylation (one-sided *t* test; ***$P < 2.2e^{-16}$). The box plots represent the distribution of H3K27ac log FC values within the range of $-0.5$ (minima) to 0.5 (maxima). The samples include four biological replicates for H3K27Ac ChIP-seq (2 wild-type and 2 MEIS KO). The center, box bounds and whiskers are defined as in C.

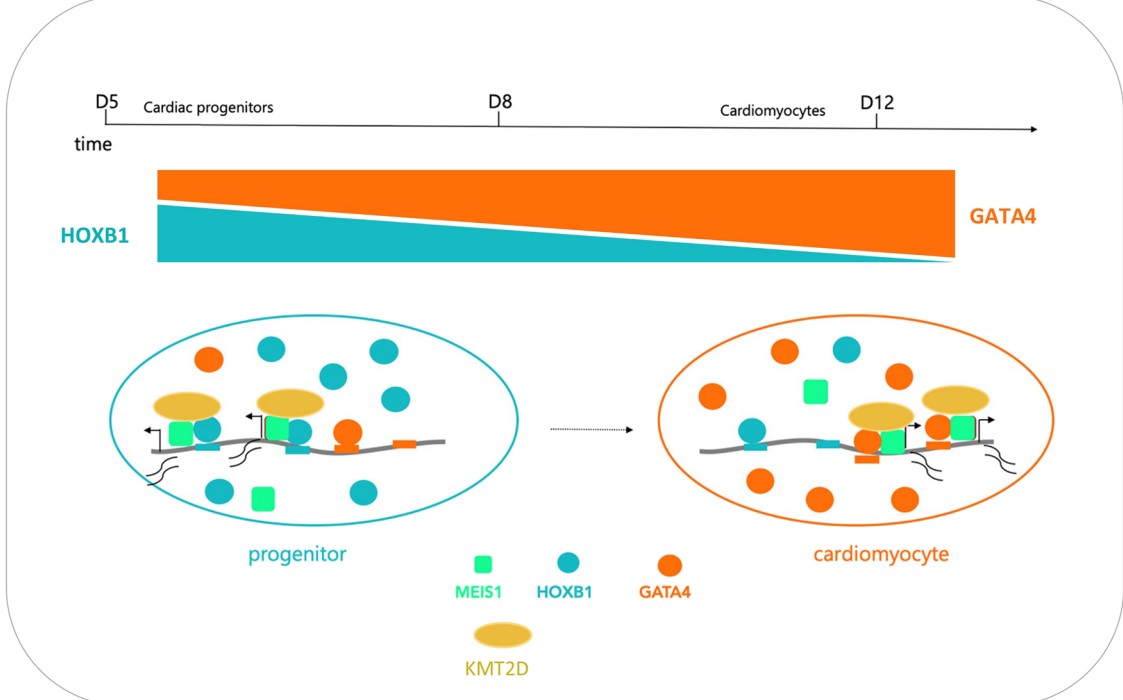

Figure 7.    Model.

Variation in TF dosage regulate the distribution of MEIS actuators at enhancers. At the progenitor stage, HOXB1 levels are high, GATA4 are low. MEIS interacts predominantly with HOX, which directs MEIS to its target enhancers. Here, MEIS1 promotes accumulation of KMT2D leading to activation of HOX target genes. As differentiation proceeds, GATA4 levels increase and HOXB1 levels decrease. This favors MEIS interaction with GATA; GATA4 directs MEIS to function at its target enhancers to drive the expression of its target genes. Combinatorial binding is dynamic and associated with the progression of cardiac differentiation: MEIS actuators fully activate an "early" cardiac progenitor program in concert with HOX and a subsequent specification module together with GATA TFs.

cardiac TFs like GATA. The relationship with HOX will be relevant in areas where HOX are expressed, notably the posterior SHF (Stefanovic et al, 2020) and Juxta-cardiac field (JCF) (Tyser et al, 2021). This restriction may help to maintain cells in a progenitor state, allowing for sufficient proliferation before committed progenitors enter terminal

differentiation. Therefore, we propose that the dynamics of MEIS combinatorial binding, influenced by changes in the dosage of its binding partners, orchestrate the progression of cardiac differentiation.

TBX5 and GATA4 show extensive binding overlap, and functionally interact to drive cardiac morphogenesis (Garg et al,

2003; Gonzalez-Teran et al, 2022b; Luna-Zurita et al, 2016). TBX TFs recognize a sequence (TGNTGACA) which largely coincides with TALE motif (TGACA), and MEIS1 is among the top TBX5 interactors in cardiac progenitors (Gonzalez-Teran et al, 2022b). The intersection of GATA4-TBX5 and MEIS1 peaks is strongly linked to cardiac GO terms, suggesting a broader regulatory framework in which MEIS may collaborate with TBX5 and GATA TFs to regulate the cardiac transcriptional program.

MEIS TFs are widely expressed and, beyond cardiac differentiation, are indispensable for the development of multiple organs and can exhibit oncogenic behavior. MEIS cooperate with tissue-restricted TFs in different contexts. For instance, in the branchial arches, the homeodomain TF HOXA2 activates its transcriptional program by enhancing MEIS TF binding to potentially lower-affinity sites throughout the genome (Amin et al, 2015). In projection neurons, MEIS2 is guided to enhancers associated with projection-neuron-specific genes by the homeodomain TF DLX5 (Dvoretskova et al, 2024). In the retina, MEIS TFs collaborate with the homeodomain TF LHX2 to activate the expression of retinal progenitor cell-specific genes (Dupacova et al, 2021). MEIS TFs cooperate with HOX and TBX5 to establish early limb development (Delgado et al, 2021), and with HOXB13 to control postnatal cardiomyocyte maturation and proliferation (Nguyen et al, 2020). These examples support the model that tissue-specific transcriptional activation results from the combined effect of ubiquitously expressed actuator TFs and lineage- and/or domain-specific TFs.

In summary, our results reveal that that combinatorial binding assembles widely expressed actuators and lineage-restricted TFs into selective, transcriptionally active complexes. Tissue-specific transcription results from ubiquitous actuators harnessing general transcriptional activators at tissue-specific enhancers, to which they directed by binding with lineage- and/or domain-specific TFs. This model offers insights into how TF synergy shapes transcriptional regulation.

# Methods

### Reagents and tools table

| Reagent/resource | Reference or source | Identifier or catalog number |
| --- | --- | --- |
| **Experimental models** | | |
| HES3 NKX2-5eGFP/w hESCs | Elliott et al, 2011 | |
| **Recombinant DNA** | | |
| p276 eSpCas9_2gRNAs_hH11 | Addgene | 164850 |
| p2attNG-H11-short | Addgene | 51546 |
| p274 EF1a_EGFP_hH11 | Addgene | 164851 |
| p2attPC | Addgene | 51547 |
| HOXB1 | GenScript | NM_002144.4 |
| pMK243 (Tet-OsTIR1-PURO) | Addgene | 72835 |
| FUdeltaGW-rtTA | Addgene | 19780 |
| pCS-kI | Addgene | 51553 |

| Reagent/resource | Reference or source | Identifier or catalog number |
| --- | --- | --- |
| pCMV-Bx | Addgene | 51552 |
| **Antibodies** | | |
| MEIS1 | Abcam | ab19867 |
| IgG Rabbit Polyclonal | Merck Millipore | 12-370 |
| rabbit polyclonal anti KMT2D | Atlas Antibodies | HPA035977 |
| FLAG (M2) | Sigma | F1804 |
| GATA4 | Proteintech | 19530-1-AP |
| GATA6 | Cell Signalling | 5851 |
| GST | Sigma | G1160 |
| HRP-conjugated anti-β-ACTIN | Sigma | A3854 |
| α-actinin | Merck | A7811 |
| WT1 | Abcam | ab89901 |
| GATA6 XP Rabbit mAb | Cell Signaling Technology | D61E4 |
| **Oligonucleotides and other sequence-based reagents** | | |
| RT_PCR | | |
| ADRB3 Fw | CTCCGAGACTCCAGACCATG | |
| ADRB3 Rev | CACAGGGTTTCGATGCTGG | |
| EOMES Fw | CAACAACACCCAGATGATAGTC | |
| EOMES Rev | GGCTCATTCAAGTCCTCCAC | |
| GATA4 Fw | CGACACCCCAATCTCGATATG | |
| GATA4 Rev | GTTGCACAGATAGTGACCCGT | |
| GATA6 Fw | TGCCAACTGTCACACCACAA | |
| GATA6 Rev | TGCTATTACCAGAGCA AGTCTTTGA | |
| HEY1 Fw | GGCAGGAGGGAAAGGTTACT | |
| HEY1 Rev | GCGTAGTTGTTGAGATGCGA | |
| HOXB1 Fw | TTCAGCAGAACTCCGGCTAT | |
| HOXB1 Rev | CCTCCGTCTCCTTCTGATTG | |
| HOXB2 Fw | TTTAGCCGTTCGCTTAGAGG | |
| HOXB2 Rev | CGGATAGCTGGAGACAGGAG | |
| KCNH7 Fw | GTGAGGCTAAGAGAACTGCG | |
| KCNH7 Rev | TCTCTTCTCACACACACTGACA | |
| LRRC4 Fw | GAATACCTGTCCAAGCTGCG | |
| LRRC4 Rev | CCCTCAAAAGCTCCCTCAGA | |
| MEIS1 Fw | GACACGGCATCTACTCGTTCA | |
| MEIS1 Rev | TGTCCAAGCCATCACCTTGC | |
| MEIS2 Fw | TGATGCAACCTCAACCCACTC | |
| MEIS2 Rev | TGTCTAAACCATCCCCTTGCTC | |
| MESP1 Fw | CTCTGTTGGAGACCTGGATG | |
| MESP1 Rev | CCTGCTTGCCTCAAAGTG | |
| MYH6 Fw | GACCAGATCATCCAGGCCAA | |
| MYH6 Rev | AACTCCGGCTTCTTGTTGGA | |
| NKX2-5 Fw | TCTATCCACGTGCCTACAGC | |

| Reagent/resource | Reference or source | Identifier or catalog number |
|---|---|---|
| NKX2-5 Rev | GTTGTCCGCCTCTGTCTTCT | |
| OCT4 Fw | CAAAGCAGAAACCCTCGTGC | |
| OCT4 Rev | CACTCGGACCACATCCTTCTC | |
| SOX2 Fw | CCCAGCAGACTTCACATGT | |
| SOX2 Rev | CCTCCCATTTCCCTCGTTTT | |
| TBXT Fw | ATCACCAGCCACTGCTTC | |
| TBXT Rev | GGGTTCCTCCATCATCTCTT | |
| TNNT2 Fw | TTCGACCTGCAGGAGAAGTT | |
| TNNT2 Rev | GCGGGTCTTGGAGACTTTCT | |
| ZNF503 Fw | CATTTTGCACCCCGAGTACC | |
| ZNF503 Rev | AACCGAGGAGAGTTTGGAGG | |
| RE-ChIP qPCR | | |
| ZNF503_KM Fw | AGCGTTTGAAGGAGGTAGCT | |
| ZNF503_KM Rev | AGTTCCACCAGTGCCCATAA | |
| HOXB1_KM Fw | CAATCTTCCATCGCCCACTG | |
| HOXB1_KM Rev | GGTGACGAATGGCTGTGTTT | |
| BAALC_KM Fw | TGACAAAGTGACTGGACCGT | |
| BAALC_KM Rev | ACCTTCACCTGTCTTTCCCA | |
| BNC1_KM Fw | CCACCTCCAGCACCATTCTA | |
| BNC1_KM Rev | GTCCTGGTGCTTGTTACTGC | |
| HEY1_KM Fw | AAAACCACTCTCTCCTGCGC | |
| HEY1_KM Rev | TCCTGTTGAAAGCATCCCCT | |
| PRDM6_KM Fw | CTTCTTTGATGGCTCACTCCTAA | |
| PRDM6_KM Rev | AGAGGTTTAATTGGTGGCTCC | |
| RSPO4_KM Fw | GTCTCTCCCTTCAACCCCAT | |
| RSPO4_KM Rev | ACATTCAGGGTTAGTGGCCA | |
| NegCo1 Fw | GAAAAACGAATACAAATCTGCAA | |
| NegCo1 Rev | CGTGATCCAAAGTAGATATTGTCA | |
| ChIP qPCR | | |
| HEY1_M Fw | AGGGGATGCTTTCAACAGGA | |
| HEY1_M Rev | TTTCTACCACGACCTGCTGA | |
| ZNF503_M Fw | GCCGCTTCCCTTTCATCTC | |
| ZNF503_M Rev | AGGCGGATTATCTTCGAGGG | |
| LRRC4_M Fw | AGGACCCATTTAGTTCTAGGAGT | |
| LRRC4_M Rev | ATCAGGGCAATGTTATGGGC | |
| TNNT2_M Fw | GACAAATGCCAGACCAAGGG | |
| TNNT2_M Rev | ACCAGAGACTCTTTGCTCCC | |
| KCNH7_M Fw | GTGAGGCTAAGAGAACTGCG | |
| KCNH7_M Rev | TCTCTTCTCACACACACTGACA | |

| Reagent/resource | Reference or source | Identifier or catalog number |
|---|---|---|
| ADRB3_M Fw | ACTCAGGACTACACATGACCA | |
| ADRB3_M Rev | GAAGTACAGCAAAGGTCATGAGA | |
| NegCo2 Fw | GGATCACGAGGTCAGGAGAT | |
| NegCo2 Rev | CGGGTTCACGCCATTCTC | |
| NegCo3 Fw | CAGCCTGAGATCAAACTGCA | |
| NegCo3 Rev | TTTGTTTGTCTGTGCCCTGC | |
| DICE clone Genotyping | | |
| HA_L | AGGATGCCTTCTATATCCTCAGC | |
| HA_R | TTAGGCCTGTGTCAACAGTTTGG | |
| Gene targeting | | |
| MEIS1 Fw | TTC ACC TTC TAC CCT CGG GA | |
| MEIS1 Rev | CTA ACG CTC TCG GGC GAA AT | |
| MEIS2 Fw | AGG CTA GTT CTT CGG GGC TT | |
| MEIS2 Rev | TCC TCA CAA CTT TTA ACT CCG TT | |
| MEIS1 | 5'-CGACGATCTACCCCATTACG GGG-3' | |
| MEIS2 | 5'-CTTCAAGGCGTCGTTGACAG CGG-3' | |
| Meis deletion primers | | |
| hMeis1attb1F (before N-terminus) | GGGGACAACTTTGTACAAAAAAGTTGGCATGGCGCAAAGGTACGACG | |
| hMeis1attb2R (after C-terminus) | GGGGACAACTTTGTACAAGAAAGTTGGGTTAcatgtagtgccactgcccc | |
| hMeis1attb2R (afterHD) | GGGGACAACTTTGTACAAGAAAGTTGGGTTActggtctatcatgggctgcacta | |
| hMeis1attb1F (beforeHD) | GGGGACAACTTTGTACAAAAAAGTTGGCgacaaaagcgtcacaaaaagc | |
| **Chemicals, enzymes, and other reagents** | | |
| KnockOut™ Serum Replacement | Gibco | 11520366 |
| Non-essential amino acids | Gibco | 11140035 |
| GlutaMAX™ | Gibco | 35050038 |
| β-mercaptoethanol | Gibco | 31350010 |
| bFGF | Miltenyi Biotech | 130-115-009 |
| TrypLE Select | Thermo Fisher Scientific | A1217701 |
| BMP4 | R&D Systems | 314-BP-050 |
| Activin A | Miltenyi Biotech | 130-115-009 |
| CHIR99021 | Selleckchem | s1263 |

| Reagent/resource | Reference or source | Identifier or catalog number |
|---|---|---|
| XAV939 | Tocris | CAYM13596-1 |
| Cas9 (Alt-R™ S.p. Cas9 Nuclease V3) | IDT | 1081058 |
| Alt-R® Cas9 Electroporation Enhancer | IDT | 1075915 |
| Human Stem Cell Nucleofector Solution 2 | Lonza | VPH-5022 |
| Q5 High-Fidelity DNA polymerase | NEB | M0491S |
| Lipofectamine™ Stem Transfection Reagent | Thermo Fisher | 15781918 |
| G418 | Invitrogen | 10131035 |
| puromycin | Invitrogen | A1113803 |
| doxycycline | Sigma-Aldrich | D3072 |
| RNeasy kit | Qiagen | 74136 |
| Quantitect SYBR green reagent | Qiagen | 204143 |
| QuantiTect SYBR Green RT-PCR Kit | Qiagen | 204245 |
| Fugene6 | Promega | E2691 |
| protease inhibitor cocktail | Roche | 11873580001 |
| Glutathione-sepharose beads | Sigma | GE17-0756-01 |
| donkey serum | Merck | S30-M |
| ProLong™ Gold Antifade Mountant | Thermo Fisher Scientific | P10144 |
| Trizol | Invitrogen | 15596026 |
| 4200 TapeStation | Agilent Technologies | |
| Illumina® Stranded mRNA Prep Ligation kit | Illumina, Inc. | |
| ATAC-Seq Kit | Active Motif | 53150 |
| Chromium Controller and Single Cell 3′ Reagent Kits v3 | 10x Genomics | |
| Matrigel | Corning | 354230 |
| C59 | Tocris | 5148 |
| mTeSR™ Plus | STEMCELL | 100-0276 |
| **Software** | | |
| Rotorgene Software 2.3.5 | Corbett Research | |
| Bowtie2 | Langmead and Salzberg, 2012 | |
| SAMtools | Li et al, 2009 | |
| MACS2 v2.2.5 | Zhang et al, 2008 | |
| Motif2Site 1.10.0 | Zarrineh et al, 2022 | |
| EdgeR: 3.28.1 | Robinson et al, 2010 | |
| DiffBind v3.4.11 | Ross-Innes et al, 2012 | |
| Homer: v4.11.1 | Heinz et al, 2010 | |

| Reagent/resource | Reference or source | Identifier or catalog number |
|---|---|---|
| STAR 2.7.2b | Dobin et al, 2013 | |
| DESeq2 1.22.2 | Love et al, 2014 | |
| scran_1.12.1 | Lun et al, 2016 | |
| 10xGenomics Cell Ranger v3.1.0 | https://www.10xgenomics.com/support/software/cell-ranger | |
| dynamicTreeCut_1.63-1 | Langfelder et al, 2008 | |
| **Other** | | |

## Maintenance of human ESC lines

HES3 *NKX2-5eGFP/w* hESCs (Elliott et al, 2011) and gene-edited derivatives were maintained on a layer of mitotically-inactivated mouse embryonic fibroblasts (MEFs) in DMEM/F12-based medium containing 20% (v/v) KnockOut™ Serum Replacement (Gibco), 100 mM non-essential amino acids (Gibco), 2 mM GlutaMAX™ (Gibco), 0.1 mM β-mercaptoethanol (Gibco) and 10 ng/ml bFGF (Miltenyi Biotech). hPSCs were passaged using TrypLE Select (Thermo Fisher Scientific) every 3–4 days. hESCs line MAN13 (Ye et al, 2017) was maintained in feeder-free culture conditions in 6-well plates pre-coated with Matrigel matrix (Corning) in mTeSR1medium (Stem Cell Technologies). Cells were passaged at 80% confluence using TrypLE Select (Thermo Fisher Scientific) every 3–4 days. The culture medium was changed every 2 days.

## hESC differentiation

HES3 *NKX2-5eGFP/w* hESC differentiations were performed in serum-free BPEL medium containing 1 μg/mL insulin using an embryoid body (EB) or monolayer format. 24 h before differentiation was induced, hESCs were plated at a density of $7 \times 10^4/cm^2$ (for EBs) or $2 \times 10^4/cm^2$ (for monolayer) in normal growth medium in six-well plates. EBs were then formed by depositing 2500 cells in 50 μl differentiation medium per well in V-bottomed 96-well plates (Greiner). The following growth factors were present for the first 3 days of differentiation: 25 ng/ml BMP4 (R&D Systems), 25 ng/mL Activin A (Miltenyi Biotech) and 1.5–1.75 μM CHIR99021 (Selleckchem). On day 3, wells were refreshed with 100 μl BPEL. On day 6, wells were refreshed with 100 μL BPEL supplemented with 100 pg/ml bFGF (Miltenyi Biotech). On day 9, wells were refreshed with 100 μL BPEL. Monolayers followed the same timing but were cultured in six-well plates in 3.5 ml of medium and 5 μM XAV939 (Tocris) was included between days 3 and 6. MAN13 cardiac monolayer differentiations were performed following a modified protocol (Lian et al, 2013) using C59 WNT inhibitor at 2 mM concentration (Tocris).

## Generation of MEIS1-2 knockout hESCs

Ribonucleoprotein (RNP) complexes of CRISPR–Cas9 were made by combining 120 pmol of crRNA:tracrRNA mix (IDT) with 67 pmol of Cas9 (Alt-R™ S.p. Cas9 Nuclease V3, IDT), final concentrations 1.3 and 1.1 μM, respectively. gRNA sequences were

*MEIS1*: ACGACGATCTACCCCATTAC and *MEIS2*: CTTCAAG GCGTCGTTGACAG. *MEIS1* and *MEIS2* were targeted sequentially. Cells ($0.5 \times 10^6$) were transfected with the RNP mix and 120 pmol Alt-R® Cas9 Electroporation Enhancer (1.3 µM final). Transfection was performed by electroporation using an Amaxa Nucleofector II combined with Human Stem Cell Nucleofector Solution 2 (Lonza, ref #: VPH-5022) and program B-16. Three days later, single cells were sorted by FACS (BD Influx) into MEF-coated 96-well plates. DNA was isolated from clones using a DNA Mini kit for PCR screening (Qiagen) using primers spanning the cut sites. PCRs were performed using Q5 High-Fidelity DNA polymerase (NEB). Homozygous mutant clones were confirmed by Sanger sequencing using primers listed in the Reagents and Tools Table. MEIS KO phenotype was confirmed by generating double MEIS1-2 KO clones in a single round in the MAN13 hESC line, using the gRNA described above.

## Generation of Tet-On-HOXB1 hESC line

HES3 *NKX2-5^eGFP/w* (Elliott et al, 2011) were genetically modified by site-specific placement of Dox-inducible *HOXB1* at the *H11* locus using DICE system (Zhu et al, 2014). DICE system was introduced into the *H11* locus of *NKX2-5^eGFP/w* using two plasmids: one carrying Cas9 nuclease and two guide RNAs (Addgene#164850) and a second one carrying a landing pad (Addgene #51546) with homology arms (HAs) for homologous recombination at the *H11* locus (Addgene #164851). The donor plasmid (Addgene #51547) carried *HOXB1* gene (GenScript NM_002144.4) under control of TET-ON promoter (Addgene 72835) and modified rtTA (Tet_On 3 G) under control of the UbC promoter (Addgene#19780). Transfections were performed in 6-well plates overnight, using Lipofectamine™ Stem Transfection Reagent (Thermo Fisher), following manufacturer instructions. For the generation of Pad_H11 cell line, 2 µg of landing pad plasmid and 1 µg Cas9-sgRNA-encoding plasmid were used for Cas9-mediated recombination. For HOXB1 integration, 1 µg was used for each of phiC31 integrase (Addgene #51553), Bxb1 integrase (Addgene#51552) and HOXB1 donor plasmids. Cells were selected with either 100 µg/ml G418 (Invitrogen) for HR or 1 µg/ml puromycin (Invitrogen) for DICE, starting 2 days after transfection and selecting cells for 4 days. Clones were isolated and selected for their ability to efficiently upregulate *HOXB1* on exposure to 1 µg/ml doxycycline (Sigma-Aldrich). Genetic modification was confirmed by genotyping, followed by sequencing using primers listed in the Reagents and Tools Table.

## Real-time RT-qPCR

Total RNA was isolated using a RNeasy kit (Qiagen) according to the manufacturer's instructions and transcripts detected in a one-step RT-qPCR reaction using 30 ng of total RNA and Quantitect SYBR green reagent (Qiagen). Data were normalized against control housekeeping genes *RPLPO* or *TUBB5*. RNA samples were run in duplicate from at least three independent experiments. RT-PCR primers are listed in the Reagents and Tools Table.

## ChIP and reChIP

MEIS1 ChIP was performed in HOXB1-overexpressing cells on d8 of cardiac differentiation as described previously (Ji et al, 2012) using MEIS1 antibody (Abcam ab19867) and IgG Rabbit Polyclonal

(Merck Millipore). Bound regions were detected by qPCR using the primers listed in the Reagents and Tools Table.

For re-ChIP assay, we used $24 \times 10^6$ of HES3 *NKX2-5^eGFP/w* cells, collected on d5 of cardiac differentiation. After the first round of immunoprecipitation with rabbit polyclonal anti KMT2D (Atlas Antibodies), the beads were washed and then incubated as described in (Ji et al, 2012) with minor modifications. Precipitated complexes were eluted twice in 55 µl Elution buffer (1×TE, 2% SDS, 10 mM DTT) containing fresh protease inhibitor cocktails for 30 min at 37 °C with shaking. 10% sample was set aside as total in-line control ChIPs. To decrease SDS concentration, volume was increased 20 times with ChIP buffer (1 mM EDTA, 0.5 mM EGTA, 4 mM Tris-HCl, pH 8.0, 100 mM NaCl, 0.1% Na-deoxycholate, 17 mM N-lauryl sarcosine) and subjected to a second round of immunoprecipitation with MEIS1 antibodies or IgG control. Following reverse cross-linking and DNA purification, bound regions were detected by quantitative PCR (qPCR), using Quantitect SYBR green PCR reagent (Qiagen) and primers listed in the Reagents and Tools Table. Results were analyzed with Rotorgene Software 2.3.5 (Corbett Research) and expressed as fold enrichment relative to IgG. Two independent experiments were performed.

## Co-precipitation experiments

Coding sequences for GATA4, GATA6, HOXB1, MEIS1b, MEIS2.1, and MEIS1 deletions were cloned into mammalian expression vectors. The primers used to generate MEIS1 deletion are listed in the Reagents and Tools Table. GATA4 has a V5 tag, HOBX1 has a FLAG tag, and MEIS1b and MEIS2.1 have a GST tag. Cells were seeded in six-well plates at 400,000 cells/well and transfected after 24 h using 1 µg of plasmid and Fugene6 (#E2691, Promega) according to the manufacturer's instructions. For experiments using GATA4, HOXB1, and MEIS1b, an empty pcDNA3 vector was used to keep DNA amounts equal in each condition. Proteins were collected in ice-cold IPLS lysis buffer including protease inhibitor cocktail (#11873580001, Roche) 48 h after transfection. Glutathione-sepharose beads (#GE17-0756-01, Sigma) were pre-washed three times with ice-cold IPLS lysis buffer, and then incubated overnight with samples on a rotating wheel at 4 °C. Beads were washed three times with ice-cold IPLS. Input and CoP samples were boiled in Laemmli buffer, run on SDS-PAGE and visualized using anti-FLAG (M2) (#F1804, Sigma), anti-GATA4 (#19530-1-AP, Proteintech), anti-GATA6 (#5851, Cell Signalling), anti-GST (G1160, Sigma), HRP-conjugated anti-β-ACTIN (#A3854, Sigma), and HRP-conjugated secondary antibodies.

## Immunocytochemistry

EBs were fixed with 10% neutral buffered formalin (Merck) for 20 min, followed by 3× PBS washes. EBs were permeabilized with 0.5% Triton X-100/PBS for 1 h and blocked for 1 h with 4% donkey serum/PBS (Merck), both at room temperature. Primary antibodies against α-actinin (1:800; Merck A7811) and WT1 (1:200; Abcam ab89901) were incubated at 4 °C in blocking buffer overnight. EBs were washed 3× with 0.05% PBS/Tween-20, followed by incubation with AlexaFluor secondary antibodies for 1 h at room temperature. EBs were subsequently washed 3X with PBS/Tween-20 and nuclei stained with 1 µg/ml Hoechst for 15 min at room temperature. EBs

were washed with PBS/0.1% BSA and mounted onto coverslips using ProLong™ Gold Antifade Mountant (Thermo Fisher Scientific). Images were acquired using SP8 inverted confocal microscope (Leica).

## Next generation sequencing and downstream analyses

### Bulk RNA-seq

RNA was isolated from whole EBs using Trizol (Invitrogen). RNA was isolated from two MEIS KO clones, both of which exhibited the same phenotypic response (lack of NKX2-5/GFP expression). These are KO1 (shown in Fig. 2A) and KO2 (shown in Fig. EV2B). For each MEIS KO clone, RNA was extracted at d5 and d7 in the course of two independent differentiation (a total of four differentiations, each including two wild-type controls). The quality and integrity of the RNA samples was assessed using a 4200 TapeStation (Agilent Technologies) and libraries generated using the Illumina® Stranded mRNA Prep Ligation kit (Illumina, Inc.) according to the manufacturer's protocol. The loaded flow cell was paired-end sequenced (76 + 76 cycles, plus indices) on an Illumina HiSeq4000 instrument. RNA-seq experiments were analyzed using Trimmomatic for trimming (Bolger et al, 2014), STAR (Dobin et al, 2013) for aligning to the human genome (hg38). DESeq2 (Love et al, 2014) was used to normalize the gene expression count values, calculate fold changes, and to detect the differential expression genes across wild-type and MEIS KO using adjusted P value < 0.05 and log FC 1 as the cutoff values. Enrichr (Kuleshov et al, 2016) was used to determine the functional enrichment of the differential expressed genes. For the analysis of genes encoding for DE TFs, human TFs were downloaded from (Lambert et al, 2018) and only TFs associated with a MEIS peak using GREAT standard association rule settings (McLean et al, 2010) were used.

### ChIP-seq

ChIPmentation assays were performed as described (Xu et al, 2020), starting from 3 to 5 × 10^6 cell for TFs and 0.5 to 1 × 10^6 for H3K27ac and using the following antibodies: MEIS1(Abcam ab19867), GATA6 XP Rabbit mAb (D61E4, Cell Signaling Technology), H3K27ac (Abcam ab4729), rabbit polyclonal KMT2D (Atlas Antibodies) and monoclonal ANTI-FLAG M2 antibody (Sigma-Aldrich). ChIP-seq experiments were performed in duplicates for every condition, with the exception of MEIS1 ChIP-seq. The intersection of MEIS1 ChIP-seq with a publicly available MEIS1 ChIP-seq in cardiac progenitors (Gonzalez-Teran et al, 2022b) showed good replication of our MEIS1 peaks (60% and 52% for MACS peaks FE > 10 and the totality of MACS peaks, respectively). The quality of replicates was compared using "run_rep_compare_bigWig_-plus_Pearson_new_params_rmOutliers.sh" command from deepTools2 (Ramírez et al, 2016). This command generated heatmaps filled by the Pearson correlation coefficient values shown in Appendix Fig. S1. ChIP-seq experiments were analyzed using Trimmomatic for trimming (Bolger et al, 2014), Bowtie2 (Langmead and Salzberg, 2012) for aligning to the human genome (hg38), SAMtools (Li et al, 2009) to remove the aligned reads with a mapping quality Q30 and MACS2 for peak calling (Zhang et al, 2008) with default narrow peak calling setting for TFs. Only high-confidence binding regions, defined as those with fold enrichment (FE) > 10, were considered for target assignments and GO analysis. GREAT standard association rule settings (McLean et al, 2010) was used to associate ChIP-seq peaks with genes and uncover events controlled by TF binding. For statistical analyses aimed at identifying trends and

patterns in the data (box plots, heatmaps) TF ChIP-seq were also analyzed with the Bioconductor package Motif2Site (Zarrineh et al, 2022), as specified in the corresponding figure legend. Motif2Site detects TF-binding sites by performing peak calling around user-provided motif sets and re-centers binding sites across replicate ChIP-seq experiments to improve the prediction accuracy. 'findMotifGenome' module of the HOMER package (Heinz et al, 2010) was used to detect de novo motif in 200-nt summit regions. For downstream analysis, no experimental wet-lab normalization methods (e.g., spike-in) were employed, which may limit the full comparability of chromatin signals across samples. Instead, the count values of each dataset were normalized using edgeR TMM normalization (Robinson et al, 2010). LogRPKM values were also calculated by edgeR. The differential peaks and log FC of all ChIP-seq experiments across wild type and MEIS KO were also calculated by edgeR using replicates. The logRPKM values were averaged across replicates and mean-centered to plot heatmaps for ATAC-seq and ChIP-seq (H3K27ac, KMT2D, GATA6) across wild type and MEIS KO. For H3K27ac and KMT2D ChIP-seq, short reads were counted in open chromatin regions from ATAC-seq. Regions of size 3000 and 1000 nucleotides for H3K27ac and KMT2D, respectively, were centered on open chromatin peaks. ggplot2 package (Wickham, 2016) was used to generate all the used plots, GenomicRanges package (Lawrence et al, 2013) was used to perform operations on genomic intervals, and eulerr was used to generate Venn diagrams (https://cran.r-project.org/web/packages/eulerr/vignettes/venn-diagrams.html).

### ATAC-seq

Cell pellet lysis, tagmentation and DNA purification were performed on 50–100 × 10^3 cells/sample using ATAC-Seq Kit (Active Motif) following the manufacturer's instructions. As above, "run_rep_compare_bigWig_-plus_Pearson_new_params_rmOutliers.sh" command from deepTools2 (Ramírez et al, 2016) was used to compare the quality of replicates (Appendix Fig. S1). ATAC-seq experiments were analyzed using Trimmomatic for trimming (Bolger et al, 2014), Bowtie2 (Langmead and Salzberg, 2012) for aligning to the human genome (hg38), SAMtools (Li et al, 2009) to remove the aligned reads with a mapping quality Q30 and MACS2 for peak calling (Zhang et al, 2008) with parameter setting for ATAC-seq peak calling. DiffBind (Ross-Innes et al, 2012) was used to re-center peaks across the called peaks and generated the count data in those peaks. EdgeR (Robinson et al, 2010) was used to normalize count values, calculate fold changes, and detect differential open chromatin regions using FDR 0.1 as cutoff value. 'findMotifGenome' module of the HOMER package was used to detect de novo motif in differential accessible regions (Heinz et al, 2010). GREAT standard association rule settings (McLean et al, 2010) was used to annotate differential accessible regions across wild type and MEIS KO.

### Single-cell RNA-seq

Differentiations were started on sequential days so the different timepoints could be collected and processed on the same day. Sample 1 was collected on day 3 and day 3.75 (50:50 mix); Samples 2, 3, and 4 were collected on days 4.75, 5.75, and 12, respectively. EBs were dissociated to single cells using TrypLE Select. Gene expression libraries were prepared from the single cells using the Chromium Controller and Single Cell 3′ Reagent Kits v3 (10x Genomics) according to the manufacturer's protocol (CG000183 Rev A). Sequencing libraries comprised standard Illumina paired-end constructs. Paired-end sequencing (26:98) was performed on the Illumina NextSeq500 platform using NextSeq500/550 High Output v2.5 (150

Cycles) reagents. The .bcl sequence data were processed for QC purposes using bcl2fastq software (v. 2.20.0.422) and the resulting .fastq files assessed using FastQC (v. 0.11.3), FastqScreen (v. 0.9.2) and FastqStrand (v. 0.0.7) prior to pre-processing with the CellRanger pipeline. Raw sequencing data were processed using the (v3.1.0). Base call (BCL) files generated by the sequencer were converted to FASTQ files using "cellranger mkfastq". The FASTQ files were mapped against the pre-built human reference package from 10X Genomics (GRCh38-3.0.0). After removing outlier cells (total read counts >40,000) 4860 cells (3509 cells from day 3 and 3.75; 5013 cells from day 4.75; 3984 cells from day 5.75; and 2124 cells from day 12) remained for downstream analysis. We filtered out genes with average UMI counts per cell below 0.01. To account for differences in sequencing depth per cell, we normalized the raw counts using a deconvolution-based method and then log-transformed with a pseudo-count of 1 added to all counts. The QC-filtered cells from the different timepoints were aggregated for downstream analysis. Highly Variable Genes (HVGs) were identified using the modelGeneVar function from scran (Lun et al, 2016), using biological components >0.5 and FDR value < 0.1. Cell cycle phase classification was performed using the cyclone function from scran and after evaluating their impact HVGs strongly associated with cell cycle were subtracted. These HVGs were then used to reduce the dimensions of the dataset using PCA and to 2D using t-SNE, where the first 14 components of the PCA were given as input. Cells were grouped into clusters using the dynamicTreeCut package (Langfelder et al, 2008). Clusters were assigned to cell types using known marker genes. Data can be accessed and explored at: https://shiny.its.manchester.ac.uk/cardiac-gene-profiler/1.

### Identification of TALE motif in developmental enhancers

TF motifs co-occurring with tissue-specific TF's motifs at tissue-specific developmental enhancers were identified using the method described in Garcia-Mora et al (Garcia-Mora et al, 2023). Tissue-specific TFs were identified using human embryonic H3K27ac ChIP-seq data and RNA-seq data (Gerrard et al, 2020; Gerrard et al, 2016). Motifs corresponding to tissue-specific TFs were obtained from the Homer motif database under the names Gata4 (GATA_GATA4), Hand2 (BHLH_HAND1), Tbx20 (TBOX_TBX20), Olig2 (BHLH_MYOD1), Hoxc9 (HOME-OBOX_Hoxc9), RFX (RFX_RFX4), Atoh1 (BHLH_ATOH1), Nr5a2 (NR_NR5A2). Coordinates of tissue-specific motifs in the human genome (hg38) were identified using scanMotifGenomeWide.pl script, from HOMER (v4.11, 10-24-2019) (Heinz et al, 2010). Using BEDtools (Quinlan and Hall, 2010), tissue-specific motif coordinates were identified in putative tissue-specific enhancers (Gerrard et al, 2020) to use as foreground. Tissue-specific motif coordinates found in random genomic regions were used as background for motif enrichment analysis. Motif enrichment analysis was performed using the findMotifGenomeWide.pl script, from HOMER using a size parameter of 200 nt. A $P$ value cutoff of $1 \times 10^{-5}$ was used and only motifs occurring in 5% or more of foreground regions in HOMER results were considered. The R package ggplot (Wickham, 2016) was used to produce bubble plots for network visualization, where negative log $P$ value determine bubble size.

### Statistical analysis

A one-sided $t$ test was conducted to compare the log2 FC of normalized count values from ATAC-seq experiments, as well as H3K27ac, GATA6, and KMT2D ChIP-seq experiments, between wild-type and MEIS KO cells at d5 of cardiac differentiation. To assess protein interactions, a Tukey HSD test was applied to

evaluate differences in the levels of GATA4 co-precipitated with MEIS1 in the presence or absence of HOXB1. Similarly, the same test was used to compare the levels of HOXB1 co-precipitated with MEIS1 in the presence or absence of GATA4.

## Data availability

All data described have been deposited in ArrayExpress with the following accession numbers: E-MTAB-14241: Replicate HOXB1 ChIP-seq. https://www.ebi.ac.uk/biostudies/arrayexpress/studies/E-MTAB-14241?key=bfb9a882-87fe-4755-b068-782084de770e. E-MTAB-14243: ChIP-seq for MEIS1; Replicate ChIP-seq for GATA6 (wild-type and MEIS KO); H3K27Ac (wild-type and MEIS KO); KMT2D (wild-type and MEIS KO). https://www.ebi.ac.uk/biostudies/arrayexpress/studies/E-MTAB-14243?key=32ff8af0-9e86-4628-bfb4-e7200871cb86. E-MTAB-14242: Replicate ATAC-seq (wild-type and MEIS KO; day 5 and day 12). https://www.ebi.ac.uk/biostudies/arrayexpress/studies/E-MTAB-14242?key=cbf29d8d-228a-4811-abf8-5cc36479304f. E-MTAB-14250: RNA-seq (wild-type and MEIS KO cardiac differentiation). https://www.ebi.ac.uk/biostudies/arrayexpress/studies/E-MTAB-14250?key=72bc2388-f3a8-4939-a20c-702c54ceeccc. E-MTAB-14248: RNA-seq (time course of hESC cardiac differentiation). https://www.ebi.ac.uk/biostudies/arrayexpress/studies/E-MTAB-14248?key=57199fd6-1f9c-435b-9d13-ef820c5ed59c. E-MTAB-14249: ScRNA-seq (time course of hESC cardiac differentiation). https://www.ebi.ac.uk/biostudies/arrayexpress/studies/E-MTAB-14249?key=340fdf5f-0ec3-462d-b943-17dff0238e59. Single-cell data can be accessed and explored at: https://shiny.its.manchester.ac.uk/cardiac-gene-profiler/1.

The source data of this paper are collected in the following database record: biostudies:S-SCDT-10_1038-S44318-025-00385-5.

## Peer review information

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

## Acknowledgements

The authors thank Andy Hayes, Leo Zeef and members of the Genomic Technologies, Bioinformatics and Bioimaging Core Facilities at the University of Manchester. The authors also thank Charles Sagerstrom and Andy Sharrocks for critical feedback on the manuscript. This work was funded by the Biotechnology and Biological Sciences Research Council (BB/T007761 to NB and BB/X016684 to NB and MJB) and the UKRI Future Leaders Fellowship (MR/T041668/1 to MJB).

## Author contributions

**Zoulfia Darieva**: Conceptualization; Resources; Supervision; Investigation; Writing—review and editing. **Peyman Zarrineh**: Conceptualization; Formal analysis; Methodology; Writing—review and editing. **Naomi Phillips**: Resources; Formal analysis; Investigation; Writing—review and editing. **Joshua Mallen**: Resources; Investigation; Writing—review and editing. **Araceli Garcia-Mora**: Formal analysis; Investigation; Writing—review and editing. **Ian Donaldson**: Data curation; Formal analysis; Writing—review and editing. **Laure Bridoux**: Resources; Investigation; Writing—review and editing. **Megan Douglas**: Investigation; Writing—review and editing. **Sara F Dias Henriques**: Resources; Writing—review and editing. **Dorothea Schulte**: Conceptualization; Resources; Writing—review and editing. **Matthew J Birket**: Conceptualization; Resources; Formal analysis; Supervision; Funding acquisition; Investigation; Writing—review and editing. **Nicoletta Bobola**: Conceptualization; Supervision; Funding acquisition; Writing—original draft; Project administration; Writing—review and editing.

Source data underlying figure panels in this paper may have individual authorship assigned. Where available, figure panel/source data authorship is listed in the following database record: biostudies:S-SCDT-10_1038-S44318-025-00385-5.

## Disclosure and competing interests statement

The authors declare no competing interests.

# Expanded View Figures

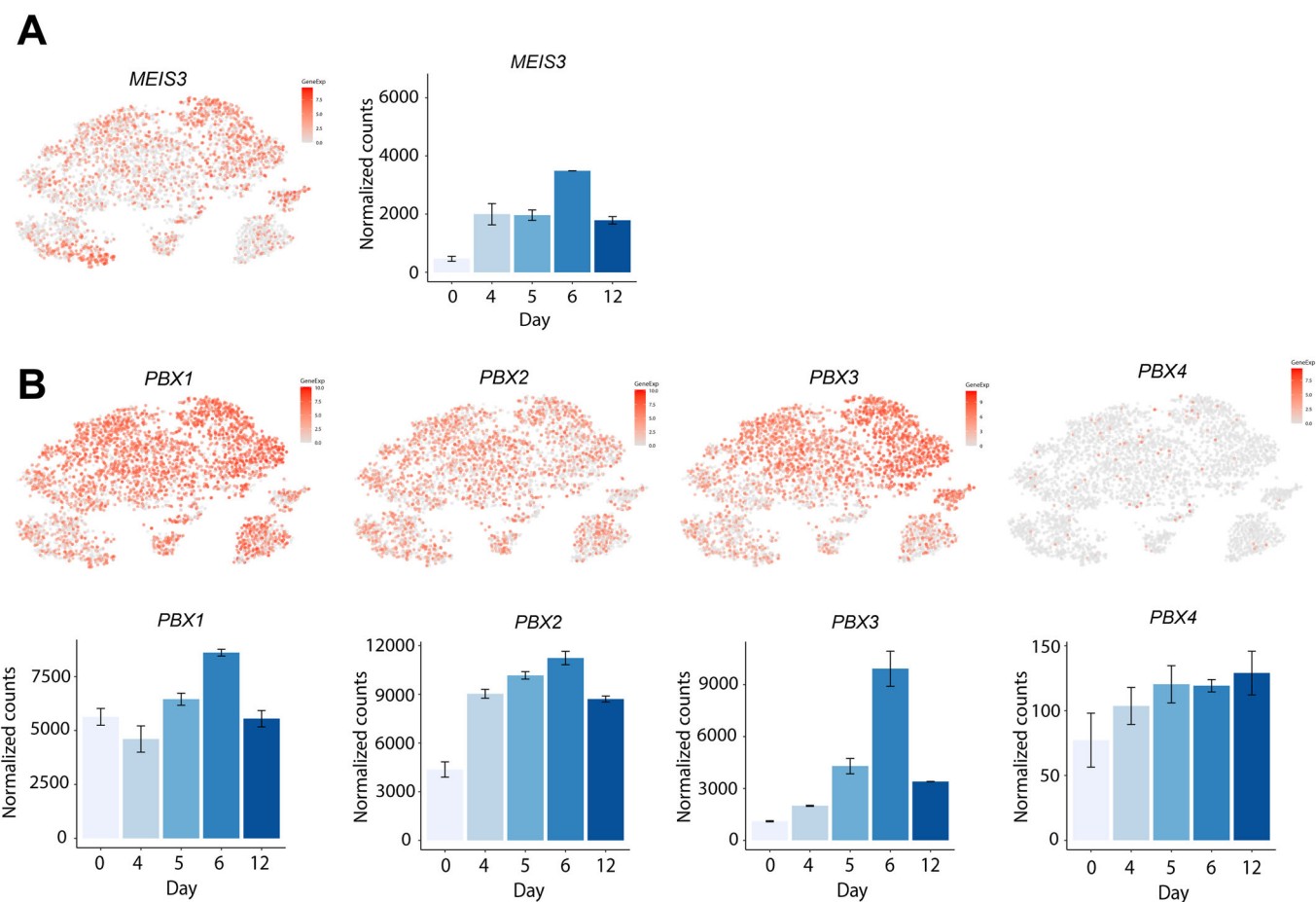

**Figure EV1. Expression of TALE genes in cardiac differentiation.**

(**A, B**) t-SNE plots showing gene expression level in the scRNA-seq cardiac differentiation time course, and corresponding bulk RNA-seq normalized counts by differentiation day, for *MEIS3* (**A**) and PBX family members (**B**). For bulk RNA-seq experiments, $n = 2$ (d0, d4, d5, d12); $n = 3$ (d6); the error bars indicate the standard deviation (SD). The scRNA-seq cell populations are annotated in Fig. 1D.

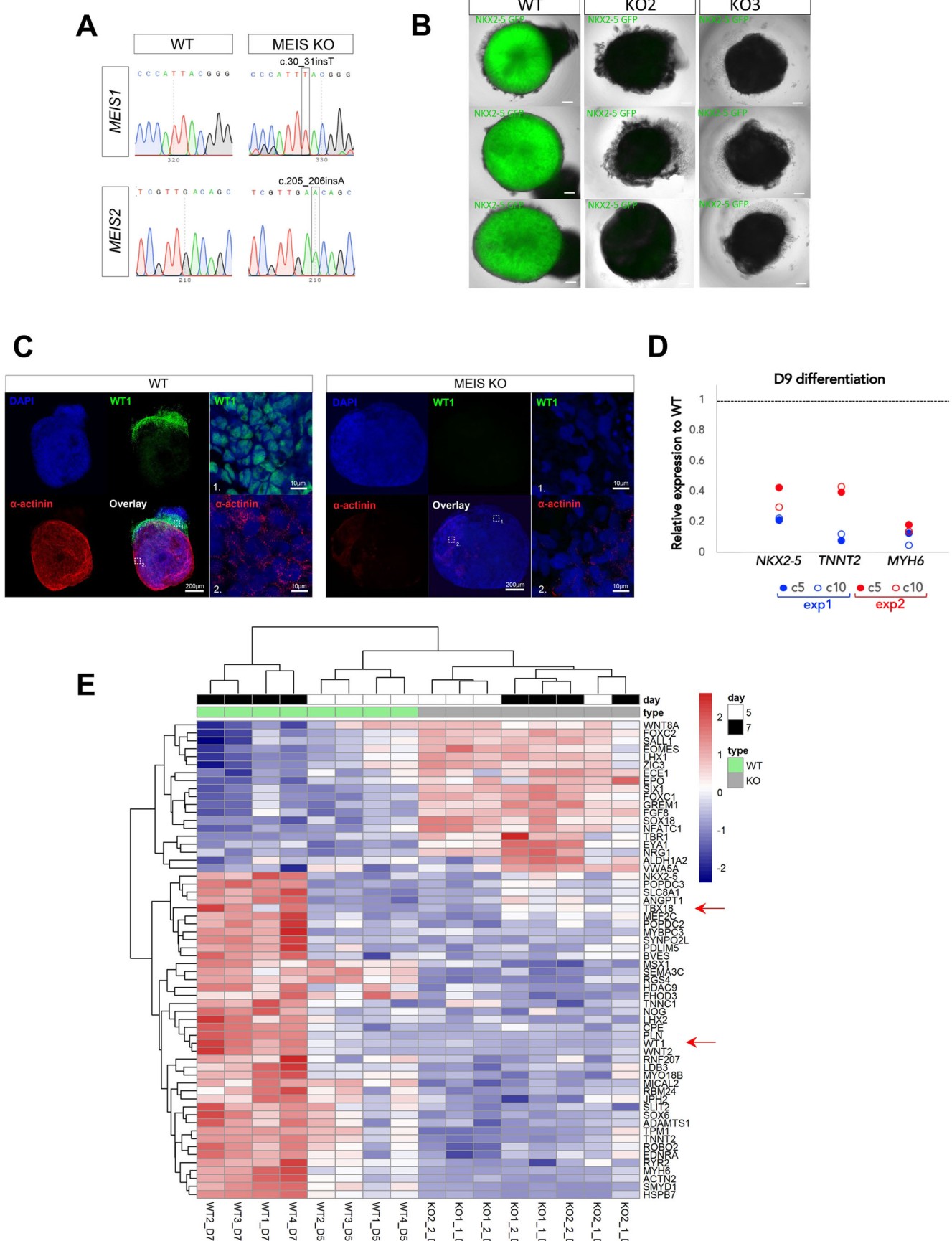

◄ **Figure EV2. Analysis of MEIS KO clones.**

(A) Sanger sequencing chromatograms showing the presence of homozygous insertion frameshift-causing mutations in *MEIS1* (c.30_31insT) and *MEIS2* (c.205_206insA) of the MEIS KO clones KO1, KO2 and KO3. The reference wild-type sequencing is also shown. (B) Live imaging of cardiac lineage reporter *NKX2-5-GFP* in wild-type and two additional MEIS KO EBs at d12 (KO2 and KO3 clones). In the same conditions, MEIS KO EBs fail to activate *NKX2-5* expression. The GFP channel is overlayed on the brightfield images. Scale bar = 200 μm. (C) Confocal microscopy image of WT1 (epicardial cells, green), α-actinin (cardiomyocytes, red) and nuclei (DAPI, blue) in day 12 EBs generated from wild-type and MEIS KO hESCs. The high magnification inserts show a sarcomere staining pattern with α-actinin and nuclear WT1, both of which are absent in the MEIS KO line. (D) Downregulation of cardiomyocyte markers in d9 MEIS KO MAN13 cells. Analysis of two independent clones (C5 and C10) with MEIS1-2 knockouts generated in a single round in the hESC line MAN13. The expression levels of *NKX2-5*, *TNNT2*, and *MYH6* were measured by qPCR in two independent differentiation experiments and are presented as percentages relative to the expression levels observed in wild-type (WT) cells. (E) Expression of cardiac mesoderm GO genes in wild-type (green) and MEIS KO (gray) cells at d5 (white) and d7(black). Top 60 genes with significant changes in expression at d7 (log FC > [1]; *P*adj <0.05) are plotted. The epicardial markers *TBX18* and *WT1* are highlighted by red arrows. Source data are available online for this figure.

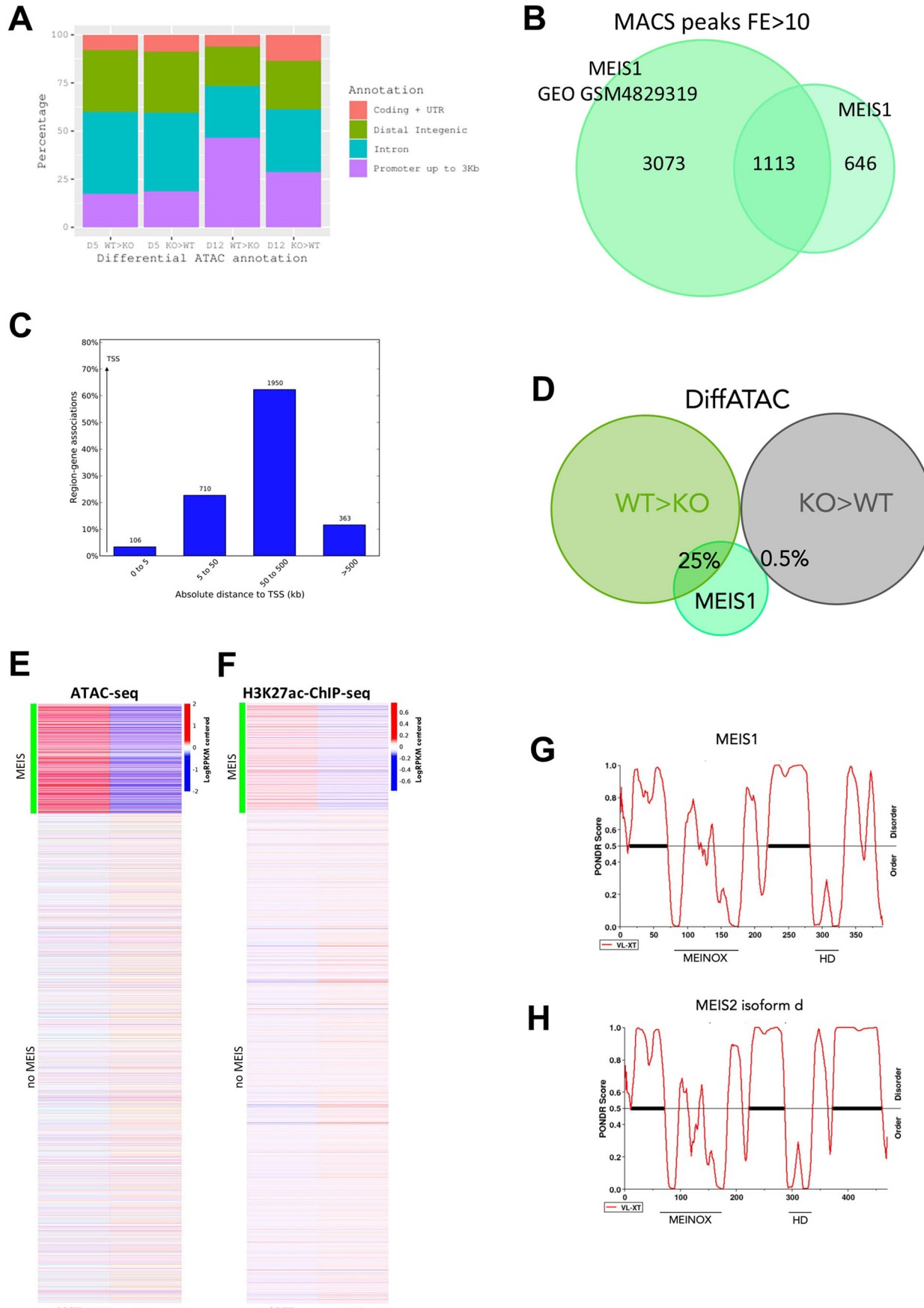

**Figure EV3. Genomic distribution of active chromatin and MEIS1 occupancy.**

(A) Genomic distribution of differential chromatin accessibility in wild-type and MEIS1-2 KO at d5 and d12. At d5, differential accessibility between wild-type and mutant is largely observed at distal regions and introns. At d12, promoters make half of the differentially accessible regions in wild-type while in the absence of MEIS1-2, distal regions and introns remain more highly accessible. (B) Comparison of MEIS1 ChIP-seq with MEIS1 ChIP-seq in d6 cardiac progenitors (GEO accession: GSM4829319). MACS peaks intersection shows that >60% of our MEIS1 peaks (FE > 10) is replicated in the GSM4829319 dataset. The overlap is 52%, when considering all MACS peaks (no enrichment cutoff). (C) Genomic distribution of top FE > 10 MEIS1 peaks. (D) Intersection of top FE > 10 MEIS1 peaks with differential ATAC-seq peaks. High-confidence MEIS1 binding overlaps almost exclusively with chromatin that is more accessible in wild-type conditions (25% of top MEIS1 peaks; $n = 446$), indicating a direct association between MEIS1 binding and accessible chromatin. In contrast, in the mutant, chromatin accessibility is independent of MEIS, with only 0.5% of top MEIS peaks ($n = 8$) intersecting ATAC peaks higher in MEIS KO. This suggests that MEIS does not directly establish repressive chromatin environments; rather, sites that become more accessible in the absence of MEIS are likely opened by upregulated TFs. (E) ATAC-seq average logRPKM values across replicates, mean-centered across wild-type and MEIS KO samples. Of 142,411 total open chromatin regions, 26,062 are occupied by MEIS1. (F) H3K27ac ChIP-seq average logRPKM values across replicates, mean-centered across wild-type and MEIS KO samples. Short reads were counted in open chromatin regions; the number of regions is as in (E). (G, H) Output of PONDR (http://www.pondr.com) (Romero et al, 1997) for MEIS1 and MEIS2 (isoform d). MEIS1-2 contain a majority of IDRs and two well-conserved structured domains, the MEINOX domain and the homeodomain (Schulte and Geerts, 2019). The disordered C-terminal domain contains MEIS1 activation domain (Bisaillon et al, 2011; Mamo et al, 2006).

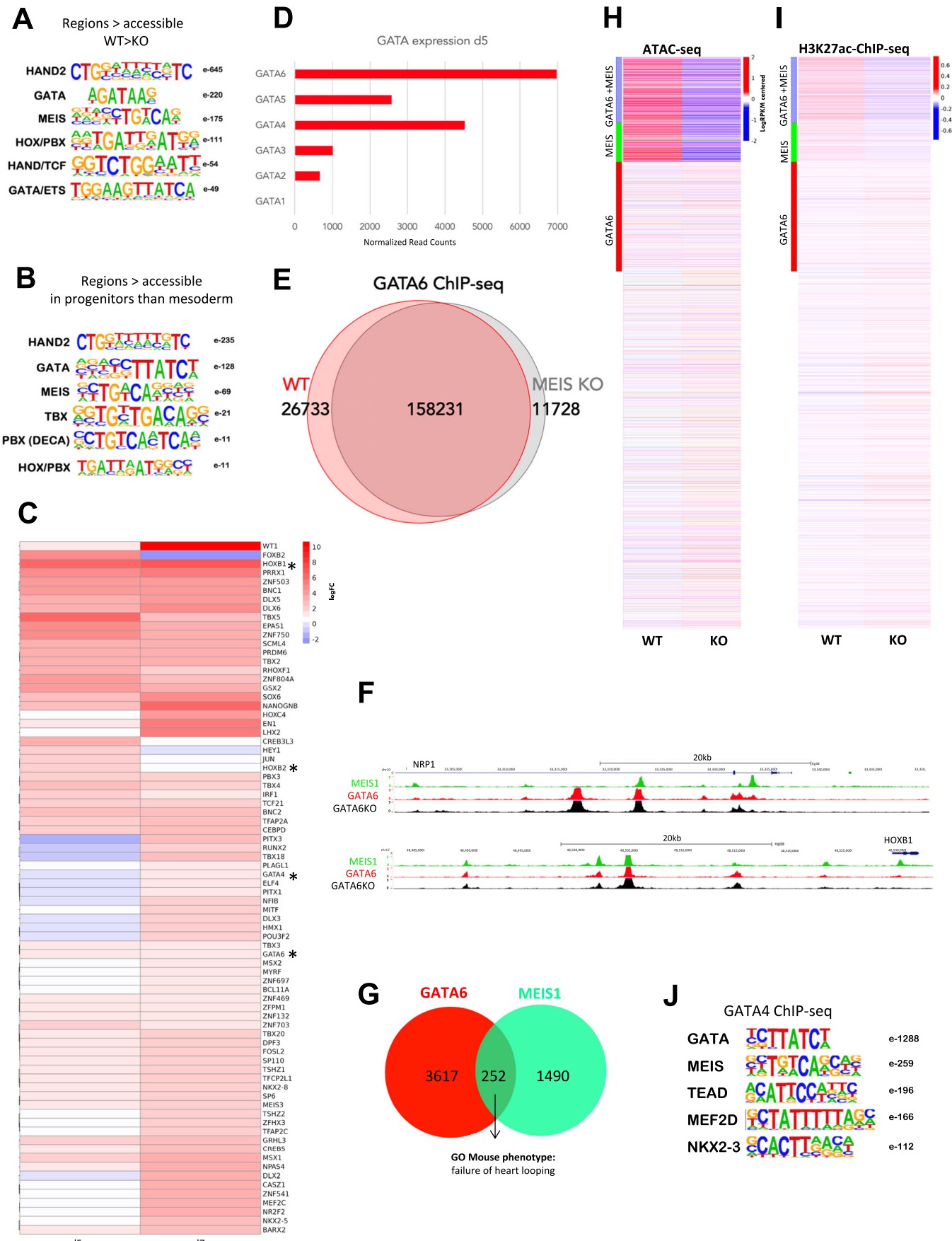

**Figure EV4.  MEIS1 and GATA6 combinatorial binding.**

(A) Top motifs, ranked by significance, identified in differential ATAC WT > KO (regions with lower accessibility in MEIS KO) using de novo motif discovery (Homer). (B) Top motifs, ranked by significance, identified in regions that increase accessibility in cardiac progenitors relative to mesoderm; data sourced from (Bertero et al, 2019b) using de novo motif discovery (Homer). (C) Differentially expressed TFs in MEIS KO. Log FC WT versus MEIS KO at d5 and d7. (D) Normalized RNA-seq counts for GATA family members in d5 cardiac progenitors. (E) Intersection of GATA6 binding in d5 WT and MEIS KO progenitors. The Venn diagram was generated using all statistically significant GATA6 peaks identified using Motif2site (Zarrineh et al, 2022). (F) UCSC tracks of MEIS1 (green) and GATA6 in wild-type and MEIS KO cells (red and black, respectively) at the *NRP1* and *HOXB1* loci. (G) Non-proportional Venn diagram generated using MEIS1 and GATA6 top peaks at d5 (FE > 10). High-confidence MEIS1-GATA6 co-occupied regions are linked to genes whose loss of function in mouse leads to heart looping failure. (H) ATAC-seq average logRPKM values across replicates, mean-centered across WT and MEIS KO samples. The total number of regions is 142411, grouped into GATA6 + MEIS1 (16315), MEIS1 only (9747) and GATA6 only (27242). (I) H3K27ac ChIP-seq average logRPKM values across replicates, mean-centered across WT and MEIS KO. Short reads were counted in open chromatin regions from ATAC-seq; the number of regions is as in (H). (J) Top motifs, ranked by significance, enriched in GATA4 ChIP-seq in d6 iPSCs-derived cardiac progenitors; data sourced from (Gonzalez-Teran et al, 2022b) using de novo motif discovery (Homer).

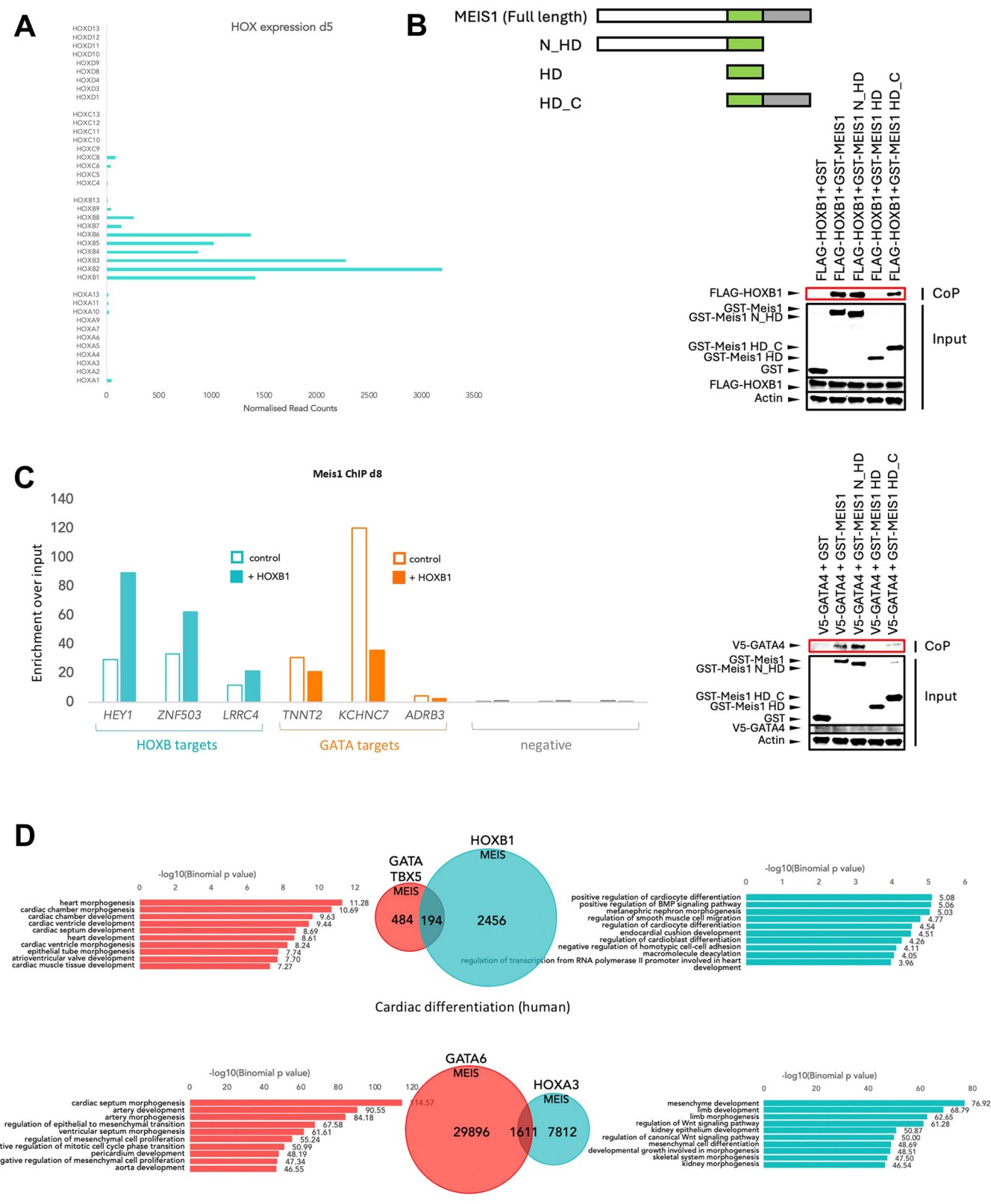

◄ **Figure EV5.  MEIS cooperate with HOX and GATA TFs.**

(A) Normalized RNA-seq counts for HOX family members in d5 cardiac progenitors. (B) Co-precipitation assays. HEK293 cells were co-transfected with expression vectors for GST alone or the following GST-tagged MEIS1 deletions: MEIS1 N_HD (including the N-terminal and HD, boxed in black), MEIS1 HD (HD only, green) and MEIS1 C_HD (including the C-terminal and HD, dark gray) and FLAG-tagged HOXB1 (top) or V5-tagged GATA4 (bottom). Protein interactions were assayed by co-precipitation on glutathione beads directed toward the GST tag and eluted proteins analyzed by western blotting to detect the presence of FLAG-HOXB1 and V5-GATA4 (red box, CoP). No binding was detected with the HD alone, possibly due to incorrect folding. Cell lysates were analyzed by western blotting prior to co-precipitation to detect protein expression (input), including ubiquitously expressed actin, used as a control. (C) One of the experiments shown in Fig. 5G, where MEIS enrichment over input is measured at HOXB1 and GATA target enhancers in d8 control and HOXB1-overexpressing cells. Negative control loci are regions with no detectable MEIS1-binding signal, located in the vicinity of *HOXC11*, *AMO3* and *SMAD6*. (D) Overlap between MEIS-HOX and MEIS-GATA co-occupied regions. To assess the full extent of the overlap, MEIS-HOX and MEIS-GATA co-bound regions were detected on a genome-wide scale using Motif2Site (Zarrineh et al, 2022). The set of GATA regions included those co-occupied by GATA6 (this paper) and GATA4-TBX5 regions (Gonzalez-Teran et al, 2022b) in cardiac progenitors. GATA-MEIS-only regions are specifically associated with cardiac differentiation GO terms, while HOX-MEIS only regions are associated with regulatory GO terms, which include regulation of cardiac differentiation as well as regulation of other processes. (D) Overlap of GATA6, MEIS and HOXA3 binding in the cardiac neural crest-populated posterior branchial arches (PBA) (Losa et al, 2017b). Similar to human cardiac differentiation, GATA6-MEIS binding is highly cardiac-specific while MEIS-HOX is linked with diverse developmental processes. The length of the bars corresponds to the binomial raw (uncorrected) *P* values calculated by GREAT using a Binomial Test.

