## [Peer Review File · The EMBO Journal]

Ubiquitous MEIS transcription factors actuate lineage-specific transcription to establish cell fate

Zoufia Darieva, Peyman Zarrineh, Naomi Phillips, Joshua Mallen, Araceli Garcia Mora, Ian Donaldson, Laure Bridoux, Megan Douglas, Sara Dias Henriques, Dorothea Schulte, Matthew Birket and Nicoletta Bobola

Corresponding author(s): Nicoletta Bobola (Nicoletta.Bobola@manchester.ac.uk) , Matthew Birket (matthew.birket@manchester.ac.uk)

Review Timeline:

Submission Date:	3rd Sep 24
Editorial Decision:	15th Oct 24
Revision Received:	9th Dec 24
Editorial Decision:	20th Jan 25
Revision Received:	30th Jan 25
Accepted:	31st Jan 25

Editor: Daniel Klimmeck

Transaction Report:

Dear Dr Bobola,

Thank you again for the submission of your manuscript (EMBOJ-2024-118929-T) to The EMBO Journal, as well as providing us with a preliminary point-by-points response to the expert concerns raised. Please accept our apologies for getting back to you with protraction due to delayed referee input, as well as detailed discussion in the editorial team. As mentioned, your study was assessed by three reviewers with expertise in cardiac development, cell fate control and transcription, whose comments are enclosed below.

As you will see from their comments, the referees acknowledge the analysis and potential interest and value of your findings. However, referee #2 also expresses major concerns regarding the conclusive support for core claims made and requests extensive additional experimentation to consolidate the work (ref#2, pts.2,3,9). Referee #3 states that more in-depth characterisation of the interaction interface of MEIS with GATA vs HOX co-factors is required (ref#3, pt.3). This expert also asks you to extend consideration of other developmental cardiac TFs (ref#3, pt.1,2). In addition, the reviewers also raise a number of issues related to the data presentation, additional controls and improved methods annotation required, as well as and overall discussion of related literature, that would need to be conclusively addressed to achieve the level of robustness and clarity needed for The EMBO Journal.

Given the overall interest stated and broader angle of your findings, we are able to invite you to revise your manuscript experimentally to address the referees' comments along the lines indicated in the preliminary response. Specifically we concur that while as such well taken, referee #2's request on a specific genetic rescue experiment (ref#2, pt.3) can be omitted in light of the additional clone data. I need to stress though that we do require strong support from the referees on a revised version of the study in order to move on to publication of the work.

Please feel free to contact me if you have any questions or need further input on the referee comments.

When submitting your revised manuscript, please carefully review the instructions below.

Please feel free to approach me any time should you have additional questions related to this.

Thank you for the opportunity to consider your work for publication.

I look forward to your revision.

Kind regards,

Daniel Klimmeck

Daniel Klimmeck, PhD
Senior Editor
The EMBO Journal

Instruction for the preparation of your revised manuscript:

- 1) a .docx formatted version of the manuscript text (including legends for main figures, EV figures and tables). Please make sure that the changes are highlighted to be clearly visible.
- 2) individual production quality figure files as .eps, .tif, .jpg (one file per figure).
- 3) a .docx formatted letter INCLUDING the reviewers' reports and your detailed point-by-point response to their comments. As part of the EMBO Press transparent editorial process, the point-by-point response is part of the Review Process File (RPF), which will be published alongside your paper.
- 4) a complete author checklist, which you can download from our author guidelines (<https://wol-prod-cdn.literatumonline.com/pb->

assets/embo-site/Author Checklist%20-%20EMBO%20J-1561436015657.xlsx). Please insert information in the checklist that is also reflected in the manuscript. The completed author checklist will also be part of the RPF.

6) It is mandatory to include a 'Data Availability' section after the Materials and Methods. Before submitting your revision, primary datasets produced in this study need to be deposited in an appropriate public database, and the accession numbers and database listed under 'Data Availability'. Please remember to provide a reviewer password if the datasets are not yet public (see <https://www.embopress.org/page/journal/14602075/authorguide#datadeposition>).

7) Our journal encourages inclusion of *data citations in the reference list* to directly cite datasets that were re-used and obtained from public databases. Data citations in the article text are distinct from normal bibliographical citations and should directly link to the database records from which the data can be accessed. In the main text, data citations are formatted as follows: "Data ref: Smith et al, 2001" or "Data ref: NCBI Sequence Read Archive PRJNA342805, 2017". In the Reference list, data citations must be labeled with "[DATASET]". A data reference must provide the database name, accession number/identifiers and a resolvable link to the landing page from which the data can be accessed at the end of the reference. Further instructions are available at .

8) At EMBO Press we ask authors to provide source data for the main and EV figures. Our source data coordinator will contact you to discuss which figure panels we would need source data for and will also provide you with helpful tips on how to upload and organize the files.

Numerical data can be provided as individual .xls or .csv files (including a tab describing the data). For 'blots' or microscopy, uncropped images should be submitted (using a zip archive or a single pdf per main figure if multiple images need to be supplied for one panel). Additional information on source data and instruction on how to label the files are available at .

9) We replaced Supplementary Information with Expanded View (EV) Figures and Tables that are collapsible/expandable online (see examples in <https://www.embopress.org/doi/10.15252/emboj.201695874>). A maximum of 5 EV Figures can be typeset. EV Figures should be cited as 'Figure EV1, Figure EV2' etc. in the text and their respective legends should be included in the main text after the legends of regular figures.

11) For data quantification: please specify the name of the statistical test used to generate error bars and P values, the number (n) of independent experiments (specify technical or biological replicates) underlying each data point and the test used to calculate p-values in each figure legend. The figure legends should contain a basic description of n, P and the test applied. Graphs must include a description of the bars and the error bars (s.d., s.e.m.).

We realize that it is difficult to revise to a specific deadline. In the interest of protecting the conceptual advance provided by the work, we recommend a revision within 3 months (13th Jan 2025). Please discuss the revision progress ahead of this time with the editor if you require more time to complete the revisions. Use the link below to submit your revision:

Referee #1:

This manuscript explores in a human cardiac differentiation system the function of MEIS1 and 2 in regulating cardiac gene expression. The work is very well executed, and the data support the conclusions. The novelty of the work is perhaps tempered by previous literature, but this is a solid and interesting piece of work that deserves publication in a widely read journal. I would only suggest that the authors present "tornado plots" of MEIS1 occupancy vs their proposed partners eg GATA, HOX, KMT2D, and histone marks, perhaps all in relationship to one another. As it is, they are numerical and pair-wise. Also the canonical MEIS motif is identical to a common non-canonical T-Box motif. It would be interesting to examine the relative occupancy of MEIS1 and TBX5 (data in Ang et al.)

Referee #2:

Darieva et al. address the role of Meis TFs in lineage specification. They observe that Meis TFs are expressed in cardiac progenitors and by CRISPR KO of Meis1 and 2 show that MEIS TFs are essential for proper cardiomyocyte formation. They suggest that Meis TFs act mainly as transcriptional activators and that they need to cooperate with lineage specific TFs to exert function. They further suggest that Gata TFs and Hox TFs compete for Meis binding and that the interaction of Meis and lineage TFs is important for proper regulation of gene expression. Mechanistically, they conclude that Meis recruits KMT2D to enhancers to activate transcription.

Meis TF are well known transcriptional co-factors. The authors discuss some contexts where this is the case in the manuscript. The idea that lineage specific TFs compete for ubiquitous transcriptional regulators is not original and true for many developmental contexts, but to my knowledge the specific interaction between Meis and Gata6 in the context of cardiac development is novel and will make an important contribution to e.g. the fields of mechanistic cardiac development and transcriptional regulation.

Overall, the experiments are performed well, they are reported in concise and clear language. However, although interesting, some of the conclusions require more experimental support. Some of the data is difficult to assess in absence of important controls or proper reporting of experimental design and/or data processing.

Major concerns:

1. Data accession numbers given in the manuscript lead to accession errors. In absence of information on replicate number for some NGS experiments and report of basic QC it is near impossible to assess the quality of the data.
2. Meis dKO hESCs fail to properly differentiate into cardiomyocytes. It is not explicitly stated how many KO cell lines were established. Looking at Fig S2, it is only one clone. If that is indeed the case, and because the derivation of the dKO cells required two rounds of clonal selection (potentially facilitating all kinds of artefacts), no conclusions can be drawn from these experiments. At least two clones, ideally using a reciprocal approach (WT Meis1 KO dKO, WT Meis2KO dKO) are required.
3. Genetic rescue experiments should be performed to establish solid cause consequence relationships.
4. Are ChIPmentation and other chromatin profiling experiments performed in a way to allow quantitative comparisons (e.g. spike ins, or other reliable normalization methods)? No specific measures are described in the methods section. If not, the authors will have to prove that their quantitative ChIP analyses are meaningful. This pertains to all comparative ChIP analysis e.g. Figs. 3E, 4D, E,...
5. Data in Fig3D could probably be better visualized by showing a chromatin heatmap of ATACpeaks, separated between Meis and non-Meis targets and K27Ac peaks separated between Meis and non-Meis targets. WT and Meis mutant data can be shown side by side.
6. Basic info and QC of ChIP-seq data is missing. It is unclear how many replicates have been performed and how consistently they behaved. This info must be added and basic QC such as a PCA should be shown.
7. Cutoff based analysis methods such as Venn diagrams bear the risk of missing important information. Visualization of ChIP-seq in multiple instances would be improved if the authors could show chromatin heatmaps. For example, they could aggregate all peaks determined in Fig 4B (Gata6 and Meis1) and split them in 3 groups. G6 only-shared-Meis1 only. Then they could plot Gata6 and Meis1 signal on these groups, both in WT and in Meis1 KOs (also enhancer marks can be plotted on these peak groups). This would make a comprehensive evaluation of the data possible and show enrichment data without relying on strict

cutoffs. This template could also be used for Fig. 6 to show the overlap with KMT2D.

8. Protein-protein interaction between Gata6 and Meis appears to be dependent on overexpression. Hence, these data should be discussed carefully. Furthermore, it is unclear whether the interaction depends on protein-protein interaction or is mediated via chromatin. Did the authors add nuclease to the co-IP to exclude a chromatin mediated co-IP?

9. It is unclear where Gata6 binds in the absence of Meis1? Fig 4D seems to show that Meis targets are less bound, but what about the fate of the Gata6-Meis double bound loci. Is the effect even stronger?

10. How reliable is the Gata6 ChIPseq data? There are close to 200k peaks called, which seems a lot for a transcription factor.

11. The sentence "While it is well established that lineage-specific TFs are essential for lineage-specific transcription, our results indicate that broadly expressed TFs are also essential." Should be toned down. The finding that broadly expressed cofactors coregulate transcription mediated by lineage specific TFs is textbook knowledge and cannot be claimed by the authors.

12. The overlap with Kdm2d seems rather low. Some statistical test should be performed to investigate whether it is significant. Regardless, the small overlap, ~1500 Meis1 targets are not co-occupied by KMT2D, suggests that the model proposed by the authors only holds for the minority of Meis targets. How are the other targets regulated? Are Meis/Kdm2d targets more affected in Meis1 KO's?

13. Are Kmt2d/Meis1 shared targets also bound by Gata6 or Gata4 (with the caveat of the Gata6 ChIP appearing to show very promiscuous binding)? This would be required for the authors' model.

Minor points:

1. To make reading easier, the authors should indicate in each plot at which stage of cardiac differentiation the analyses were performed. This is not always obvious.

2. FigS5C is not properly labelled.

3. Are DEGs defined indeed using a p value instead of an adj, p value cutoff. If so, why?

Referee #3:

The manuscript by Darieva et al. investigates the role of MEIS transcription factors in cardiac lineage specification using a human ESC model. MEIS 1/2 are co-expressed in cardiac progenitors and MEIS2 is also expressed in differentiated cardiomyocytes and the epicardium. In cardiac enhancers, MEIS binding sites are near those of important lineage-restricted TF such as GATA, TBX, and HAND factors. The CRISPR-Cas9-mediated ablation of MEIS 1/2 results in impaired myocardial and epicardial differentiation. MEIS are associated with chromatin regions implicated in cardiovascular development. The authors specifically investigate the binding of MEIS binds to regulatory chromatin adjacent to HOXB1/B2 and GATA4/6 binding sites. MEIS and GATA cooperatively bind and inducing higher chromatin accessibility. Moreover, the authors show that the two factors display physical interaction using an overexpression system. Interestingly, HOXB1/B2 can compete for MEIS, and forced expression of HOXB1 cause an impairment of cardiac differentiation. Like GATA, HOXB1/B2 physically interacts with MEIS proteins. Finally, the authors demonstrate that gene activation involves the recruitment of the histone methyltransferase KMTD2.

This is fine piece of work, which provides novel insight into how MEIS are modulating cardiac gene expression during progenitor development and cardiomyocyte differentiation. The manuscript provides a lot of excellent data.

Minor concerns

1. I am missing an attempt to link these findings to cardiac development and regeneration. It would be good to discuss the role of MEIS in progenitor recruitment of both first and second heart field lineages. Thus, data from Jovanovic et al. (PMID: 32804075) suggest involvement of Hoxa1 and Hoxb1 in controlling myocardial differentiation in second heart field lineage recruitment. Would your findings suggest an involvement of MEIS in this context? Moreover, you demonstrate an involvement of MEIS in regulating TBX5 expression. TBX5 is a first heart field-specific cardiac transcription factor. Does it mean that MEIS might be involved in both first and second heart field lineage specification?

2. Given that your data suggest an involvement also into epicardial development. Do we know whether MEIS also targets TBX18 or WT1, two key TF involved in epicardial lineage specification?

3. How do you think MEIS can bind to the many different TF? Can you map the interaction domain involved in making physical contact with GATA and HOX proteins?

4. MEIS also has been involved in cardiac regeneration (PMID: 32499640) through an interaction with Hoxb13. I believe it would be good to cite this paper to widen the implications of your work.

5. Likewise, there is evidence of an involvement of MEIS and PBX in congenital heart malformations and I believe it is worthwhile to make such a connection in your discussion.

Please cite PMID: 18723445

6. Finally, MEIS factors might also be involved in controlling the maintenance of myocardial differentiation in the adult myocardium as recent data suggest PMID: 35777912

7. Previously, MEIS and NKX2.5 have been defined to target a hybrid binding site on the POPDC2 promoter (Ref. 46 in your manuscript), a gene that is strongly expressed in the heart and thus apart of your model of MEIS binding sites adjacent to

lineage-restricted TF binding site, another mode of action may be that MEIS is activating cardiac target genes in the early progenitor stage and subsequently is substituted by NKX2.5 binding.

8. On page 5, it is described that MEIS modulates the expression of several cardiac transcription factors, however HAND TFs which are mentioned in the manuscript are a notable exception and are not listed in Fig S4C.

9. You mention on page 5 that NRP1 is a critical factor for cardiac development and differentiation yet genetic evidence suggests mainly a role in controlling cardiac metabolism and playing a role in vascular development.

10. I think a summary slide visualizing your main findings and depicting a model of the biological activity of MEIS is currently missing.

We thank the Reviewers for their constructive feedback, which has helped us address key areas and improve the clarity and strength of the manuscript.

Below is a detailed response to each of the points raised by the Reviewers.

Reviewer 1

I would only suggest that the authors present "tornado plots" of MEIS1 occupancy vs their proposed partners eg GATA, HOX, KMT2D, and histone marks, perhaps all in relationship to one another. As it is, they are numerical and pair-wise.

We appreciate the reviewer's suggestion to present the relationships within the datasets. As an alternative to the tornado plot, we opted for heatmaps to visualize the data. This approach effectively captures the relationships between MEIS binding, its partners, open chromatin, and acetylation in a comprehensive manner, addressing the underlying objectives of both Reviewers 1-2's suggestions.

We have added heatmaps in Fig S3, Fig 4, Fig S4 and Fig 6 (please see replies to points 5 and 7 raised by reviewer 2 for more detailed explanation).

The canonical MEIS motif is identical to a common non-canonical T-Box motif. It would be interesting to examine the relative occupancy of MEIS1 and TBX5 (data in Ang et al.).

This is an interesting observation, and we have addressed the similarity between the MEIS1 and TBX5 recognition motifs in the discussion section. In the current manuscript (Fig EV5D), we highlight the intersection of TBX5 and GATA4 with MEIS1, showing that this set of peaks is highly significant for cardiac development and differentiation based on the GO analysis of associated genes.

To address the reviewer's query, we conducted an additional intersection analysis of MEIS1 and TBX5 binding sites using Motif2Site. This analysis utilized the TBX5 dataset from Gonzalez-Teran et al., which also served as the source for the GATA4 and MEIS1 datasets used in our study. The results of this analysis are now included in the Appendix (within this document). However, since the manuscript primarily focuses on the HOX and GATA model of competition for MEIS1 binding, we believe that including the MEIS1-TBX5 overlap (without GATA TFs), without conducting a deeper investigation to contextualize this interaction within the existing model, could detract from the central narrative and reduce overall clarity. We have revised the discussion section slightly (see highlighted text) to include mention of the intersection between TBX5 and GATA4 with MEIS1.

Reviewer 2

Major concerns:

1. Data accession numbers given in the manuscript lead to accession errors. In absence of information on replicate number for some NGS experiments and report of basic QC it is near impossible to assess the quality of the data.

We apologize for the missing links that caused the accession errors encountered by the reviewer, they should have been included alongside the accession numbers. The submissions can be accessed using the links below, with the data becoming available for download upon publication.

Please see also point 6 for information on replicates, QC and related changes in the manuscript.

E-MTAB-14243 (ChIP-seq). This submission contains all the ChIP-seq, except HOXB1:

<https://www.ebi.ac.uk/biostudies/arrayexpress/studies/E-MTAB-14243?key=32ff8af0-9e86-4628-bfb4-e7200871cb86>

E-MTAB-14241(HOX ChIP-seq). This submission contains HOXB1 ChIP-seq:

<https://www.ebi.ac.uk/biostudies/arrayexpress/studies/E-MTAB-14241?key=bfb9a882-87fe-4755-b068-782084de770e>

E-MTAB-14242 (ATAC-seq). This submission contains ATAC-seq experiments:

<https://www.ebi.ac.uk/biostudies/arrayexpress/studies/E-MTAB-14242?key=cbf29d8d-228a-4811-abf8-5cc36479304f>

E-MTAB-14249 (Single cell RNA-seq):

<https://www.ebi.ac.uk/biostudies/arrayexpress/studies/E-MTAB-14249?key=340fdf5f-0ec3-462d-b943-17dff0238e59>

E-MTAB-14248 Bulk RNA-seq timecourse:

<https://www.ebi.ac.uk/biostudies/arrayexpress/studies/E-MTAB-14248?key=57199fd6-1f9c-435b-9d13-ef820c5ed59c>

E-MTAB-14250 (MEIS WT and KO RNA-seq):

<https://www.ebi.ac.uk/biostudies/arrayexpress/studies/E-MTAB-14250?key=72bc2388-f3a8-4939-a20c-702c54ceccc>

Additional information about replicates has been added to the ChIP-seq method section (highlighted text).

2. Meis dKO hESCs fail to properly differentiate into cardiomyocytes. It is not explicitly stated how many KO cell lines were established. Looking at Fig S2, it is only one clone. If that is indeed the case, and because the derivation of the dKO cells required two rounds of clonal selection (potentially facilitating all kinds of artefacts), no conclusions can be drawn from these experiments. At least two clones, ideally using a reciprocal approach (WT Δ Meis1 KO Δ dKO, WT Δ Meis2KO Δ dKO) are required.

We analyzed three dKO clones in the human embryonic stem cell (hESC) HES3, KO1, KO2 and KO3. They consistently demonstrated lack of NKX2-5/GFP expression and no

observable contraction at day 14 (d14). Two of these clones, KO1 and KO2 were analysed by RNA-seq and displayed consistent downregulation of cardiac genes, as presented in Fig 2E. Except for one replicate analysed at day 7 (d7), all other replicates showed a clear and reproducible downregulation of cardiac and mesoderm markers, confirming the robustness of the phenotype. Following these results, chromatin experiments were conducted on one representative clone (KO1).

KO1 is shown in the main figure (Fig. 2A); we have added additional images showing KO2 and KO3 phenotype to Fig EV2 (EV2B) and clarified in the text that RNA-seq was performed using two independent clones, KO1 and KO2 (highlighted in the results and methods). The labels have been replaced accordingly in the heatmap in Fig 2E.

Recently, we extended our analysis by knocking out MEIS1-2 in another hESC line, MAN13. This time, we employed a different approach and generated dKO clones in a single round, which ensures clones are truly independent. Analysis of two independent clones parallels the findings in HES3: each clone showed a consistent lack of contraction as well as failure to upregulate cardiac markers. The results of this analysis are presented in Fig EV2D.

3. Genetic rescue experiments should be performed to establish solid cause consequence relationships.

As outlined in point 2, we have consistently observed the dKO phenotype - absence of contraction and failure to upregulate key cardiac markers- across multiple (five) clones, generated using different experimental approaches (MEIS2 KO followed by MEIS1KO in HES3, or simultaneous dKO in MAN13). This high level of reproducibility suggests that a genetic rescue experiment may not substantially alter our conclusions or provide significant additional insights. For these reasons, we have opted not to include a genetic rescue experiment in the revised manuscript.

4. Are ChIPmentation and other chromatin profiling experiments performed in a way to allow quantitative comparisons (e.g. spike ins, or other reliable normalization methods)? No specific measures are described in the methods section. If not, the authors will have to prove that their quantitative ChIP analyses are meaningful. This pertains to all comparative ChIP analysis e.g. Figs. 3E, 4D, E,...

For our quantitative comparisons, we applied standard normalization methods during the computational analysis of ChIP-seq data, edgeR and TMM normalization. These are widely recognized methodologies that are commonly used to ensure reliable comparisons between different ChIP-seq experiments (e.g., see Ross-Innes et al, <https://doi.org/10.1038/nature10730>). We have clarified this in the Methods section (see highlighted text). Additionally, our experiments were conducted using a reproducible pipeline with standardized parameters, including consistent cell numbers and chromatin quantities prior to precipitation.

5. Data in Fig3D could probably be better visualized by showing a chromatin heatmap of ATACpeaks, separated between Meis and non-Meis targets and K27Ac peaks separated between Meis and non-Meis targets. WT and Meis mutant data can be shown side by side.

We have provided heatmaps of ATAC peaks and K27Ac peaks, sorted by overlapping MEIS1 and not overlapping MEIS1 in both WT and MEIS KO, as requested.

While the heatmaps are visually more striking than boxplots, the latter offer quantitative information along with statistical significance, which we believe is essential to present. Therefore, we kept the boxplots in the main figure (Fig. 3DE) and added the chromatin heatmaps to the corresponding expanded view figure (Fig. EV3DE).

6. Basic info and QC of ChIP-seq data is missing. It is unclear how many replicates have been performed and how consistently they behaved. This info must be added and basic QC such as a PCA should be shown.

All genomic experiments (ChIP-seq and ATAC-seq) were conducted in duplicate, except MEIS1 ChIP-seq (which was compared to publicly available data, more information below). Additional information about replicates has been added to the ChIP-seq method section (highlighted text).

We have used “run_rep_compare_bigWig_plus_Pearson_new_params_rmOutliers.sh” command from deepTools2 (Ramirez et al 2016) to compare the quality of replicates. This command generated heatmaps filled by the Pearson correlation coefficient values.

We used Pearson correlation to compare the following:

ATAC-seq experiments performed at the same stage:

1. 4 x ATAC-seq d5 (2 WT and 2 MEIS KO)
2. 4 x ATAC-seq at d12 (2 WT and 2 MEIS KO)

ChIP-seq experiments performed at the same stage and with the same antibody:

3. 4 x H3K27Ac ChIP-seq at d5 (2 WT and 2 MEIS KO)
4. 4 x GATA6 ChIP-seq at d5 (2 WT and 2 MEIS KO)
5. 4 x KMT2D ChIP-seq at d5 (2 WT and 2 MEIS KO)
6. 2 x HOXB1 ChIP-seq at d8

Pearson correlations are shown in the Appendix (this document). The replicated experiments showed consistent results overall. However for HOXB1, one replicate was of lower quality. Importantly, the use of HOXB1 ChIP-seq in the manuscript was more limited compared to other datasets. While other ChIP-seq experiments were used for quantitative comparisons of binding between wild-type and KO, as well as for identifying overlap with MEIS binding, accessibility, and acetylation, HOXB1 ChIP-seq was primarily employed to confirm HOX binding to predicted HOX-MEIS enhancers. This limited scope is explicitly outlined in the Results section: “We first selected HOXB1-MEIS target enhancers based on specific criteria, including overlap with a top MEIS1 peak, increased accessibility in wild-type cells, presence of at least one HOX-PBX consensus motif, and association with a DE gene in MEIS KO cells. Subsequently, we validated HOXB1 binding to these regions using ChIP-seq (Fig 5F).”

We have not included the QC in our figures, as we believe this information is redundant given that all datasets will soon be publicly accessible.

MEIS1 ChIP-seq: We compared our MEIS1 ChIP-seq to MEIS1 ChIP-seq in day 6 cardiac progenitors (Gonzalez-Teran et al. 2022; GEO accession: GSM4829319). The majority of our MEIS1 peaks were replicated in the Gonzalez-Teran dataset, confirming the robustness of our results. Intersection of MACS peaks shows that > 60% of our MEIS1 peaks FE >10 is replicated in Gonzalez-Teran MEIS1 ChIP-seq. When considering all MACS peaks (no cutoff) 52% of our MEIS1 peaks are replicated in Gonzalez-Teran MEIS1 ChIP-seq (Venn diagrams are provided in the appendix). Both our and Gonzalez-Teran datasets were generated using the same antibody (ab19867, Abcam). The overlap is now mentioned in the result section (highlighted) together with a Venn diagram provided in Fig EV3B).

7. Cutoff based analysis methods such as Venn diagrams bear the risk of missing important

information. Visualization of ChIP-seq in multiple instances would be improved if the authors could show chromatin heatmaps. For example, they could aggregate all peaks determined in Fig 4B (Gata6 and Meis1) and split them in 3 groups. G6 only-shared-Meis1 only. Then they could plot Gata6 and Meis1 signal on these groups, both in WT and in Meis1 KOs (also enhancer marks can be plotted on these peak groups). This would make a comprehensive evaluation of the data possible and show enrichment data without relying on strict cutoffs.

We have generated three chromatin heatmaps as suggested by the reviewer.

1. We have plotted GATA6 binding levels (log2RPKM) in all GATA6 peaks in WT and MEIS KO, split into GATA6 only and MEIS1-GATA6. We have not included MEIS1 only peaks because MEIS1 is not present in MEIS KO (we did not attempt to map MEIS1 binding in MEIS KO). The heatmap has replaced the original Venn diagram in Fig 4B.
2. We have created two additional heatmaps for ATAC-seq and H3K27Ac (log2RPKM). Here we have sorted ATAC and K27Ac peaks into three groups: overlapping GATA6 only; overlapping GATA6 and MEIS1; overlapping MEIS1 only, and plotted ATAC-seq and H3K27Ac signals for each group. The new heatmaps are shown in Fig EV4H and Fig EV4I

This template could also be used for Fig. 6 to show the overlap with KMT2D.

We have used the same template to generate heatmaps with KMT2D peaks, shown in Fig 6D.

8. Protein-protein interaction between Gata6 and Meis appears to be dependent on overexpression. Hence, these data should be discussed carefully. Furthermore, it is unclear whether the interaction depends on protein-protein interaction or is mediated via chromatin. Did the authors add nuclease to the co-IP to exclude a chromatin mediated co-IP?

The caveat of overexpression for the interpretation of the data has been mentioned in the discussion (highlighted text).

We have not used nucleases in the coprecipitation assays. However, the methodology used would not be able to detect chromatin-mediated interaction for two main reasons. Detection of chromatin-mediated interactions relies on crosslinking; these assays do not include a crosslinking step to fix proteins on DNA and they are unlikely to detect any transient, chromatin mediated interaction. In addition, the lysates used for analysis do not contain DNA. In these assays, DNA is not sonicated and is discarded with the insoluble phase.

9. It is unclear where Gata6 binds in the absence of Meis1? Fig 4D seems to show that Meis targets are less bound, but what about the fate of the Gata6-Meis double bound loci. Is the effect even stronger?

The absence of MEIS1-2 affects GATA6 binding at a quantitative level, rather than qualitative. In terms of genomic location, GATA6 binds to similar regions with and without MEIS1-2 (see Fig EV4F, and also more specific examples in EV4E and 4F).

Loss of MEIS1-2 has a quantitative effect on GATA6 binding, which is mainly observed at GATA6-MEIS1 double bound loci (relative to GATA6 only). The plot in Figure 4C displays changes in GATA6 binding levels at two sets of regions: those bound by GATA6 alone (originally labelled as 'no MEIS1') and those where both GATA6 and MEIS1 are bound together (originally labelled as 'MEIS1'). When comparing GATA6 binding levels in wildtype relative to MEIS KO cells, there is no significant change in GATA6 binding at sites where GATA6 binds without MEIS1 (no MEIS1). However, a significant change in GATA6 binding

levels (= higher binding levels measured in WT cells), is observed at MEIS1-GATA6 co-occupied regions.

As the plot displays GATA6 binding levels, MEIS1-only regions (no GATA6 binding) are not shown in this plot.

To improve the clarity of the plot, we have updated the labels: "no MEIS" has been replaced with "GATA6," and "MEIS1" has been changed to "GATA6-MEIS1" to better reflect the results.

10. How reliable is the Gata6 ChIPseq data? There are close to 200k peaks called, which seems a lot for a transcription factor.

We performed four GATA6 ChIP-seq (two in WT cells and two in MEIS KO cells). Pearson correlation analysis shows good agreement between experiments (see also point 6 and the appendix). The type of antibody (and its performance) used in ChIP-seq is a key factor influencing the number of statistically significant peaks observed across different TF ChIP-seq experiments. Beyond antibody performance, two observations may explain the broad occupancy of GATA6 in this context. First, GATA6 peaks tend to occur in clusters. In open chromatin regions bound by GATA6, the average number of GATA6 binding sites (i.e. peaks) per region is 2.7. Second, GATA6 is expressed early in cardiac differentiation (prior to MEIS1 and MEIS2) and is observed already in mesoderm before commitment to cardiac mesoderm, as well as in cardiac mesoderm and endothelial cells. This suggests that GATA6 enables cardiac as well as alternative fates, which may be reflected in its broader chromatin occupancy. It is possible that the widespread genomic occupancy of GATA6, observed in early cardiac progenitors, still reflects the broader fate potential associated with GATA6 TFs.

The sentence "While it is well established that lineage-specific TFs are essential for lineage-specific transcription, our results indicate that broadly expressed TFs are also essential." Should be toned down. The finding that broadly expressed cofactors coregulate transcription mediated by lineage specific TFs is textbook knowledge and cannot be claimed by the authors.

We completely agree with the reviewer that the role of broadly expressed cofactors in coregulating transcription mediated by lineage-specific transcription factors is well-established and considered textbook knowledge. However, we would like to clarify that according to the standard definition in transcriptional regulation (see for example the recent review by Kim and Wysocka, doi: 10.1016/j.molcel.2022.12.032), cofactors are proteins that do not directly bind to DNA but instead assemble on DNA-bound transcriptional regulators. In contrast, transcription factors, like MEIS proteins, are DNA-binding proteins, and therefore, based on this definition, MEIS proteins are not classified as cofactors.

We acknowledge the broad involvement of ubiquitous cofactors in tissue-specific transcriptional regulation. However, we believe that examples of ubiquitous DNA-binding transcription factors, such as MEIS, that are essential for tissue-specific transcription are rare. One other example of a ubiquitous DNA-binding transcription factor with a similar role is SP1, as highlighted by Gilmour et al. (DOI: 10.1242/dev.106054). Given this context, we believe our findings provide new insight into the previously underappreciated role of ubiquitous transcription factors, like MEIS, in tissue-specific transcriptional regulation.

11. The overlap with Kdm2d seems rather low. Some statistical test should be performed to investigate whether it is significant. Regardless, the small overlap, ~1500 Meis1 targets are not co-occupied by KMT2D, suggests that the model proposed by the authors only holds for the minority of Meis targets. How are the other targets regulated? Are Meis/Kdm2d targets more affected in Meis1 KOs?

We appreciate the reviewer's attention to this issue and agree with the reviewer that Figure 6A is potentially misleading. Its main purpose is to show that, while the top KMT2D peaks are generally associated with metabolic processes, the top KMT2D peaks overlapping MEIS1 binding are specifically linked to cardiovascular processes. Figure 6A displays only the overlap of top (FE > 10) KMT2D and MEIS1 peaks; however, the overlap between KMT2D and MEIS1 binding is much broader. We have clarified that this analysis focuses exclusively on the top fraction of KMT2D and MEIS1 peaks, both in the figure legend and in the results section (highlighted text).

To address the reviewer's concern, we have added a heatmap to Figure 6D, showing KMT2D binding signals across all open chromatin regions (grouped into categories based on MEIS1, MEIS1 and GATA6, and GATA6 alone). This new visualization clearly demonstrates that KMT2D has a much broader overlap with MEIS1 target regions than what is shown in Figure 6A and highlights that MEIS1/KMT2D target regions are more significantly affected in the MEIS knockout. This analysis also partially addresses the reviewer's subsequent question (point 12) - whether KMT2D occupancy occurs on regions co-bound by MEIS1 and GATA6.

12. Are Kmt2d/Meis1 shared targets also bound by Gata6 or Gata4 (with the caveat of the Gata6 ChIP appearing to show very promiscuous binding)? This would be required for the authors' model.

Yes. The heatmap in Figure 6D illustrates the binding signal of KMT2D across all open chromatin regions, categorized into groups bound by MEIS1, MEIS1 and GATA6, and GATA6 alone. Additionally, to address the reviewer's question, we generated pie charts of KMT2D differential binding between WT and MEIS KO cells (shown in Fig 6F and Fig 6G). Most KMT2D regions with decreased signals in MEIS KO cells overlap MEIS1 binding, either alone (~20%) or in combination with GATA4, GATA6, or both (~45.5%). Conversely, only a small fraction of KMT2D regions with increased signals in MEIS KO cells overlap MEIS1 binding alone (~4%) or in combination with GATA4, GATA6, or both (<15%).

These results support the model in which MEIS1/2 recruit KMT2D to lineage-specific, GATA-occupied enhancers, facilitating activation of the cardiac gene expression program.

Minor points:

1. To make reading easier, the authors should indicate in each plot at which stage of cardiac differentiation the analyses were performed. This is not always obvious.

The stage of cardiac differentiation has been specified in all the figure legends.

2. FigS5C is not properly labelled.

The missing C has been added to the figure

3. Are DEGs defined indeed using a p value instead of an adj, p value cutoff. If so, why?

Thank you for pointing out this mistake. The cutoff of <0.05 used to define DEGs was indeed based on the adjusted p-value, not the p-value. We have corrected this in the text.

Reviewer 3

Minor concerns

1. I am missing an attempt to link these findings to cardiac development and regeneration. It would be good to discuss the role of MEIS in progenitor recruitment of both first and second

heart field lineages. Thus, data from Jovanovic et al. (PMID: 32804075) suggest involvement of Hoxa1 and Hoxb1 in controlling myocardial differentiation in second heart field lineage recruitment. Would your findings suggest an involvement of MEIS in this context?

We believe that MEIS will have a role in cardiac progenitors across lineages by collaborating with GATA factors to promote differentiation. The relationship with HOX will be relevant in areas where HOX are expressed, notably the pSHF (<https://doi.org/10.7554/eLife.55124>) and Juxta-cardiac field (JCF) (<https://doi.org/10.1126/science.abb2986>). Based on our data and model, in these fields the Hox expression will act to buffer MEIS-GATA collaboration and restrain progenitor differentiation and instead encourage self-renewal. This is consistent with the temporal lag in differentiation reported in these two lineages (<https://doi.org/10.7554/eLife.30668>) and JCF (<https://doi.org/10.1016/j.cell.2023.01.001>). In the aSHF and FHF, which are largely free of Hox expression, MEIS-GATA collaboration will not be restricted by HOX.

We have specified in the discussion that the relationship with HOX will be restricted to those areas where HOX are expressed (highlighted in the discussion).

Moreover, you demonstrate an involvement of MEIS in regulating TBX5 expression. TBX5 is a first heart field-specific cardiac transcription factor. Does it mean that MEIS might be involved in both first and second heart field lineage specification?

As noted above, we believe that MEIS will play a role in all cardiac progenitor lineages. As noted by the reviewer, TBX5 expression is mainly restricted to the FHF. Restricted TBX5 activation maybe achieved by MEIS in collaboration with other, lineage-specific, partner TFs. Alternatively, different signaling environments of cardiac progenitors may explain TBX5 restricted expression pattern.

2. Given that your data suggest an involvement also into epicardial development. Do we know whether MEIS also targets TBX18 or WT1, two key TF involved in epicardial lineage specification?

TBX18 and WT1 are significantly downregulated at d7 (not d5), therefore they do not appear in the heatmap shown in current Fig 2E. This heatmap built using DEGs at d5, that are associated with cardiac mesoderm.

We have built a new heatmap using DEGs at d7, that are associated with cardiac mesoderm (top 60 DEGs). The heatmap, which displays changes in *TBX18* and *WT1* is now shown in Fig S2D (now Fig EV2D).

3. How do you think MEIS can bind to the many different TF? Can you map the interaction domain involved in making physical contact with GATA and HOX proteins?

To investigate MEIS1 interactions with GATA4 and HOXB1, we tested three MEIS1 deletions: (1) MEIS1 Δ C-terminus (= N-terminal + homeodomain), (2) MEIS1 Δ N-terminus (= homeodomain + C-terminal), and (3) MEIS1 HD (homeodomain only).

HOXB1 and GATA4 primarily interact with the N-terminal, including the homeodomain, of MEIS1: MEIS1 lacking the C-terminus co-precipitated GATA4 and HOXB1 comparably to full-length MEIS1, while MEIS1 lacking the N-terminus showed reduced interaction with HOXB1 and minimal interaction with GATA4. Neither protein interacted with the HD alone. This suggests that HOXB1 contacts MEIS1 at more than one site and/or that the isolated HD may not fold correctly. Data are presented in current Figure EV5B.

This data provides additional support to the model of competition between HOX and GATA proteins for MEIS binding. However, considering the likely role of DNA in facilitating HOX and GATA cooperation with MEIS - evidenced by the co-occurrence of their cognate binding

sites - and in line with reviewer 2's comment (point 8) regarding the use of overexpression experiments to study these interactions, a comprehensive characterization of HOX, GATA and MEIS binding interfaces would require the resolution of an X-ray crystal structure of the HOX-MEIS and GATA-MEIS DNA-binding domains in complex with their cognate DNA sites. This level of structural analysis is beyond the scope of the present manuscript.

4. MEIS also has been involved in cardiac regeneration (PMID: 32499640) through an interaction with Hoxb13. I believe it would be good to cite this paper to widen the implications of your work.

We thank the reviewer for this and the following reference suggestions (point 4, 5, 6).

The reference has been added to the discussion, where it is cited a) as an additional example of MEIS1 cooperation with domain-restricted TFs and b) in support for MEIS1 role in controlling cardiomyocyte proliferation.

5. Likewise, there is evidence of an involvement of MEIS and PBX in congenital heart malformations and I believe it is worthwhile to make such a connection in your discussion. Please cite PMID: 18723445

The involvement of MEIS and PBX in congenital heart malformations has been mentioned in the discussion and the reference added.

6. Finally, MEIS factors might also be involved in controlling the maintenance of myocardial differentiation in the adult myocardium as recent data suggest PMID: 35777912

The role of MEIS in postnatal cardiomyocytes has been mentioned in the discussion and the reference added.

7. Previously, MEIS and NKX2.5 have been defined to target a hybrid binding site on the POPDC2 promoter (Ref. 46 in your manuscript), a gene that is strongly expressed in the heart and thus apart of your model of MEIS binding sites adjacent to lineage-restricted TF binding site, another mode of action may be that MEIS is activating cardiac target genes in the early progenitor stage and subsequently is substituted by NKX2.5 binding.

This is certainly a possibility. We have found no compelling evidence of NKX2-5 competition with MEIS, at least not in cardiac progenitors: we did not detect enrichment of NKX2-5 motifs in MEIS peaks or in chromatin regions whose accessibility depends on MEIS1-2. However, it is possible that NKX2-5 will compete with MEIS at later stages.

8. On page 5, it is described that MEIS modulates the expression of several cardiac transcription factors, however HAND TFs which are mentioned in the manuscript are a notable exception and are not listed in Fig S4C.

We thank the reviewer for raising this point. Fig S4C (now Fig EV4C) exclusively shows TFs whose expression is regulated by MEIS1-2. Expression of genes encoding for HAND TFs is not significantly affected at these stages, and therefore they are not included in the figure. To avoid misinterpretation, we have adjusted the reference to Fig EV4C to appear earlier in the paragraph, as the figure exclusively shows TFs regulated by MEIS1-2, as follows:

“HOXB1, HOXB2, GATA4 and GATA6, whose recognition sites are enriched in MEIS1 ChIP-seq and in regions that gain accessibility in cardiac progenitors, are also activated by MEIS1-2 (Fig EV4C), with HAND TFs being a notable exception.” Referring to Fig EV4C at the end of the paragraph was erroneously hinting at HAND2 data being shown in the figure.

9. You mention on page 5 that NRP1 is a critical factor for cardiac development and differentiation, yet genetic evidence suggests mainly a role in controlling cardiac metabolism and playing a role in vascular development.

The Reviewer is correct. In addition NRP1 knock out mice fail to septate the outflow tract of the heart and display persistent truncus arteriosus.

10. I think a summary slide visualizing your main findings and depicting a model of the biological activity of MEIS is currently missing.

A model depicting the role of MEIS in cardiac differentiation is presented in Fig 7.

Appendix

Pearson correlation of ATAC-seq and ChIP-seq experiments

Intersection of MEIS1 ChIP-seq with GEO GSM4829319 dataset

MEIS1 GEO GSM4829319 (Gonzalez-Teran et al. 2022)

Intersection of MEIS1 and TBX5 ChIP-seq, analyzed with Motif2Site

Dear Dr Bobola,

Thank you for submitting your revised manuscript (EMBOJ-2024-118929R) to The EMBO Journal, as well for your patience with our response. Your amended study was sent back to the three referees for their scientific re-evaluation, and we have received detailed comments from all of them, which I enclose below. As you will see, the experts state that the work has been substantially enhanced by the revisions and they are now broadly in favour of publication, pending minor revision.

Thus, we are pleased to inform you that your manuscript has been accepted in principle for publication in The EMBO Journal.

Please carefully consider the remaining minor points raised by reviewer #2 regarding data presentation and interpretation of the findings, adjusting and complementing the manuscript text and related figures where appropriate.

We also now need you to take care of a number of issues related to formatting and data display as detailed below, which should be addressed at re-submission.

Please contact me at any time if you have additional questions related to below points.

Thank you for giving us the chance to consider your manuscript for The EMBO Journal. I look forward to your final revision.

Again, please contact me at any time if you need any help or have further questions.

Best regards,

Daniel Klimmeck

>> Section order should be corrected as follows: title page with complete author information, abstract, keywords, introduction, results, discussion, methods, data availability section, acknowledgements, disclosure and competing interests statement, references, main figure legends, tables, expanded figure legends.
i.e. Methods should be after Discussion.

>> Figure callouts: Please ensure that the figures and panels are called out correctly and in sequential order. Currently, there is a callout for a Fig S4F.

>> Add a separate 'Statistical analysis' section, detailing the algorithms and statistical tests applied.

>> Source data: source data should be uploaded as one (zipped) file per figure.

>> Dataset EV legends: Tables EV1 and EV2 should be renamed Dataset EV1 and EV2 and both should have the legend added to the top of the page.

>> Provide a completed Author Checklist: please enter corresponding author name, journal and manuscript number. select a response for 'Design >> Sample definition'.

>> Please recheck references for the bioRxiv entries Zarrineh et al. (2022) and Garcia-Mora et al. (2023); update the citation if in the meantime published as regular article.

>> Consider additional changes and comments from our production team as indicated below:

- DAS:

1. Please note that the specific URLs for E-MTAB-14241, E-MTAB-14243, E-MTAB-14242, E-MTAB-14250, E-MTAB-14248, E-MTAB-14249 datasets are not provided in the data availability statement.

>>> now in AD 16.1.24

2. Please note that reviewer access codes for E-MTAB-14241, E-MTAB-14243, E-MTAB-14242, E-MTAB-14250, E-MTAB-14248, E-MTAB-14249 datasets are not provided in the data availability statement.

>>> data now accessible AD 16.12.24

- Figure legends:

1. Please note that the exact p values are not provided in the legends of figures 3D, E; 4C, E; 6C, H.

2. Please indicate the statistical test used for data analysis in the legends of figures 1A, 4F, 5E, 6A, EV5 D.

3. Please note that the box plots need to be defined in terms of minima, maxima, centre, bounds of box and whiskers, and percentile in the legends of figures 3D, E; 4C, E; 6C, H.

4. Please note that information related to n is missing in the legends of figures 1J, 2B, 3D, E; 4C, E, F; 5E, 6B, C, H; EV1 A, B.

5. Please note that the error bars are not defined in the legends of figures 1J, 2B, 3B, C, D, E; 5E, 6B, EV1 A, B.

6. Please note that the scale bar needs to be defined for figures 2A, EV2 B.

7. Please note that scale bar and its definition are missing for figures 5A.

Please use the link below to submit your revision:

Referee #1:

The authors have carefully and completely addressed my comments.

Referee #2:

Most comments have been addressed. A few points remain open.

1. I appreciate the clarification on replicate numbers and numbers of independent clones. The number of clones and replicates for each experiment should be indicated in the text or Figure legend to avoid confusion.

2. Quantitative comparisons of ChIP data without normalization is difficult. This remains a limitation, as the updated methods section indicates that no spike in or other normalization was performed to account for potential global changes in chromatin occupancy that would not be detectable in standard approaches. My point about comparability of chromatin signals across samples remains open and this limitation must at least be clearly and openly addressed in the manuscript.

3. I believe there was a misunderstanding about the term "chromatin heatmaps". What I meant was plots usually used to visualize ChIPseq data. That show a heatmap of signal of all peaks. Centered at the peak center +/- a reasonable distance, often ~5kb. E.g. The R tool deeptools generates these plots. They have the advantage to show relatively unprocessed signal across all called peaks. Heatmaps as shown are suboptimal and not state of the art for visualizing ChIPseq data (for reference: https://deeptools.readthedocs.io/en/develop/content/example_gallery.html).

I suggest adding these plots to visualize ChIP and ATACseq data as suggested in my original comments.

4. Basic QC such as PCAs (or replicates shown in heatmaps) showing all replicates of ChIPseq and ATACseq data is still missing.

5. Fig 4B now indicates that Gata6/Meis double targets are (significantly?) more Gata6 bound compared to non-targets. I think this is important information.

6. How certain are the authors that peak calling identified meaningful peaks for Gata6. As indicated before, the number of peaks seems very high. The fact that Meis Gata6 double targets are so much more bound appears to indicate that possibly some of the Gata6 peaks called might be rather noisy. Some more detailed investigation into this is required. Some clarity might come from suggested chromatin heatmaps which plot all signal (Meis, Gata6, KMT2D, chromatin marks, ATACseq) on e.g. Meis only, Gata6/Meis and Gata6 only peaks. These plots should be added and are necessary to allow proper evaluation and interpretation of these data. I assume that Rev1's request for "tornado plots" addressed a similar point.
7. I suggest indicating that "KO" refers to "MEIS-KO" for clarity.
8. The colour code in Fig 4C is redundant and can be deleted. It still says no MEIS and therefore is rather confusing.

Referee #3:

The revised manuscript has satisfactorily addressed my comments and concerns.

All editorial and formatting issues were resolved by the authors.

Dear Dr Bobola,

Thank you for submitting the revised version of your manuscript. I have now evaluated your amended manuscript and concluded that the remaining minor concerns have been sufficiently addressed.

I am thus pleased to inform you that your manuscript has been accepted for publication in the EMBO Journal.

Related, I would like to hereby ask your consent on keeping the referee figures included in this file.

On a different note, I would like to alert you that EMBO Press offers a format for a video-synopsis of work published with us, which essentially is a short, author-generated film explaining the core findings in hand drawings, and, as we believe, can be very useful to increase visibility of the work. Please see the following link for representative examples and their integration into the article web page:

<https://www.embopress.org/doi/full/10.15252/emj.2019103932>

Finally, we have noted that the submitted version of your article is also posted on the preprint platform bioRxiv. We would appreciate if you could alert bioRxiv on the acceptance of this manuscript at The EMBO Journal in order to allow for an update of the entry status. Thank you in advance!

Best regards,

Daniel Klimmeck

Daniel Klimmeck, PhD
Senior Editor
The EMBO Journal
EMBO

Postfach 1022-40
Meyerhofstrasse 1
D-69117 Heidelberg
contact@embojournal.org
